# Dynamic mechanochemical feedback between curved membranes and BAR protein self-organization

Anabel-Lise Le Roux [1,8] ✉, Caterina Tozzi [2,8], Nikhil Walani[2], Xarxa Quiroga [1], Dobryna Zalvidea[1], Xavier Trepat[1,3,4,5], Margarita Staykova [6], Marino Arroyo [1,2,7] ✉ & Pere Roca-Cusachs [1,3] ✉

In many physiological situations, BAR proteins reshape membranes with pre-existing curvature (templates), contributing to essential cellular processes. However, the mechanism and the biological implications of this reshaping process remain unclear. Here we show, both experimentally and through modelling, that BAR proteins reshape low curvature membrane templates through a mechanochemical phase transition. This phenomenon depends on initial template shape and involves the co-existence and progressive transition between distinct local states in terms of molecular organization (protein arrangement and density) and membrane shape (template size and spherical versus cylindrical curvature). Further, we demonstrate in cells that this phenomenon enables a mechanotransduction mode, in which cellular stretch leads to the mechanical formation of membrane templates, which are then reshaped into tubules by BAR proteins. Our results demonstrate the interplay between membrane mechanics and BAR protein molecular organization, integrating curvature sensing and generation in a comprehensive framework with implications for cell mechanical responses.

[1] Institute for Bioengineering of Catalonia (IBEC), the Barcelona Institute of Technology (BIST), 08028 Barcelona, Spain. [2] Universitat Politècnica de Catalunya (UPC), Campus Nord, Carrer de Jordi Girona, 1, 3, 08034 Barcelona, Spain. [3] Universitat de Barcelona, 08036 Barcelona, Spain. [4] Institució Catalana de Recerca i Estudis Avançats (ICREA), Passeig de Lluís Companys, 23, 08010 Barcelona, Spain. [5] Centro de Investigación Biomédica en Red de Cáncer (CIBERONC), 08028 Barcelona, Spain. [6] Department of Physics, University of Durham, Durham, UK. [7] Centre Internacional de Mètodes Numèrics en Enginyeria (CIMNE), 08034 Barcelona, Spain. [8] These authors contributed equally: Anabel-Lise Le Roux, Caterina Tozzi. ✉email: aleroux@ibecbarcelona.eu; marino.arroyo@upc.edu; rocacusachs@ub.edu

ue to the curved shape and membrane binding of Bin/
Amphiphysin/Rvs (BAR) domains, proteins containing
such domains have the interesting ability to reshape
membranes. Furthermore, because of their elongated shape, they
can align along a preferred direction, adopting a nematic orga-
nization that impinges anisotropic curvature on the membrane.
In equilibrium and for high concentrations, this leads to the
generation of membrane tubes with high curvatures, comparable
to BAR intrinsic curvatures (of the order of $10^1$ nm). For instance,
incubation of small vesicles with a high concentration of BAR
proteins leads to highly curved tubes covered by a dense protein
scaffold where the elongated molecules are nematically
arranged[1,2]. In a different system, GUVs with sufficiently high
bound protein density rapidly expel thin protein-rich tubes in a
tension-dependent manner[3]. On thin membrane tubes pulled out
of giant unilamellar vesicles (GUVs), BAR proteins can also
change the radius of the tube and the force required to hold it[4].
Beyond this well-known paradigm of thin tubes in equilibrium, in
many physiological situations, BAR proteins dynamically interact
with pre-existing curved membrane templates. Such templates
can include for instance invaginations caused by nanoscale
topographical features on the cell substrate[5], mechanical folds[6,7],
or endocytic structures[8,9]. Due to their affinity for curved mem-
branes, BAR proteins are thus bound to sense and reshape such
templates in ways that are important in physiological processes[10]
like endocytosis[11], the build-up of caveolar structures[12,13], the
maintenance of cell polarity[14], and the modulation of actin
polymerization[15]. However, how BAR proteins reshape mem-
branes with initial curvatures that can be well below their
intrinsic values, with which dynamics, how this depends on initial
membrane shape, and what are the implications, remains
unknown.

To address this issue, we developed a versatile experimental
system combined with theoretical and computational modeling to
study the dynamic reshaping of cellular-like membrane structures
of a broad range of shapes and sizes. In commonly used systems
such as tube pulling assays[10] or curved substrates[16,17], imposed
curvature creates tensed curved structures. In cells, such tensed
structures (created either from nanoscale curved topographies or
from actin pulling on the membrane) can for instance recruit
N-BAR proteins and enhance endocytosis[5] or trigger the
recruitment of effectors related to actin polymerization[15,18].
However, both in vitro and in cells tensed structures prevent
extensive shape remodeling. In contrast, here we use a different
physiologically relevant signal in the form of stretch and release
cycles. In our system, we create curved membrane features off a
supported lipid bilayer (SLB) by applying a successive lateral
stretch and compression. As previously shown in SLBs[6] and in
cells[7], this leads to the storage of excess membrane area in free-
standing, low tension, easily reshaped protrusions of tubular or
spherical shape. In contrast with tubes pulled out of GUVs, where
a tip force and tension are required to stabilize their shape, in our
system tubes are stabilized osmotically without a pulling force. To
assess the effect of tension, we also control osmolarity to generate
tensed spherical caps off the SLB[6]. These protrusions emerging
from a flat SLB can serve as model system for membrane tem-
plates such as endocytic buds, or osmotically/mechanically-
induced structures.

## Results

### Experimental and theoretical framework.
Experimentally, we
used the liposome deposition method to form a fluorescently labeled
SLB on top of a thin extensible polydimethylsiloxane (PDMS)
membrane. To this end, an electron microscopy grid was deposited
on top of the PDMS membrane before plasma cleaning, which

activated only the uncovered PDMS areas[19]. An easily identifiable
hexagonal pattern was obtained (Fig. 1a), with a fluid SLB formed
inside the hexagon (Supplementary Fig. 1) while a lipid monolayer
was formed outside. The membrane was placed inside a stretching
device previously described[7] and mounted on a spinning disk con-
focal microscope (Fig. 1a). At initial state, the fluid bilayer contained
brighter signals coming from non-fused liposomes (Supplementary
Fig. 2a). This patterned SLB (pSLB) was then uniformly and iso-
tropically stretched for 120 s (until 5–8% strain), slowly enough to
allow liposome incorporation in the fluid bilayer, thereby ensuring
membrane integrity (as happens in a cellular membrane through
lipid reserve incorporation[7]). After 120 s, stretch was slowly released
during 300 s to a completely relaxed state, and lateral compression
led to the formation of highly curved lipid structure in the shape of
either lipid buds or lipid tubes (Fig. 1b, Supplementary Fig. 2a and
Supplementary Movies 1 and 2). We note that our system is
diffraction-limited and not amenable to electron microscopy, and we
could thus not measure tube diameter.

As a BAR protein, we used the commonly used model of
Amphiphysin[4,20–23], an N-BAR protein binding lipid bilayers of
positive curvature (invaginations). We assessed protein activity by
measuring the diameter of Amphiphysin-reshaped tubes via
transmission electron microscopy in the sucrose loading vesicle
assay[24]. Consistent with the literature, we found diameters of
~25 nm (Supplementary Fig. 2b). We used a pSLB composed of
negatively charged lipids necessary for Amphiphysin binding[25],
including 1,2-dioleoyl-sn-glycero-3-phosphate (DOPA) for which
Amphiphysin has a specific affinity[26]. In experiments, we injected
fluorescently labeled Amphiphysin in the bulk solution on top of
the previously described compressed pSLB (Fig. 1b), and
monitored the fluorescence signal from both the pSLB and
Amphiphysin (in different channels). Once injected, the protein
clearly bound the curved lipid buds and tubes (Fig. 1c, left) and,
after further adsorption from the bulk, it started to reshape them
into geometrically heterogeneous structures with coexistence of
small spherical and tubular features (tube-sphere complexes,
Fig. 1c, right). We then carried out several controls. First, we
monitored the tubes in absence of protein injection. In this case, a
fraction of tubes spontaneously detached, and non-detached tubes
did not undergo a progressive elongation as observed in presence
of Amphiphysin. Instead, tubes were stable for some minutes and
then progressively shortened, widened, and transformed into a
structure of spherical shape, presumably due to enclosed fluid
and/or excess membrane reorganization within the system (Fig. 1d
and Supplementary Movie 3). Eventually, these spherical struc-
tures became immobile on top of the pSLB after 10–20 min.
Second, we performed the same experiment by injecting
fluorescent Neutravidin instead of Amphiphysin. Neutravidin
did not specifically bind to the tube, and the same tube-to-bud
relaxation was observed as in absence of injection (Supplementary
Fig. 2c and Supplementary Movie 4). Third, we injected
Neutravidin and monitored its binding to a biotinylated pSLB.
We observed tube to bud relaxation as in the control, but some
longer-lived tubes immobilized in the plane of the bilayer,
providing a clear visualization of their cylindrical shape
(Supplementary Fig. 2d. and Supplementary Movie 5). Tube to
bilayer attachment is likely due to the tetrameric character of
Neutravidin, which can therefore crosslink tubes to the surround-
ing flat bilayer. Beside this effect, we observed a similar process of
relaxation from tubes to buds, clearly distinct from the tube-
sphere complexes occurring in presence of Amphiphysin. Finally,
we monitored the effect of Amphiphysin in non-stretched
membranes, which were therefore devoid of pre-existing mem-
brane structures. In this case, membrane reshaping only occurred
if Amphiphysin concentration was increased above 5 μM in the
bulk, which merely consisted in the formation of bright/dark spots

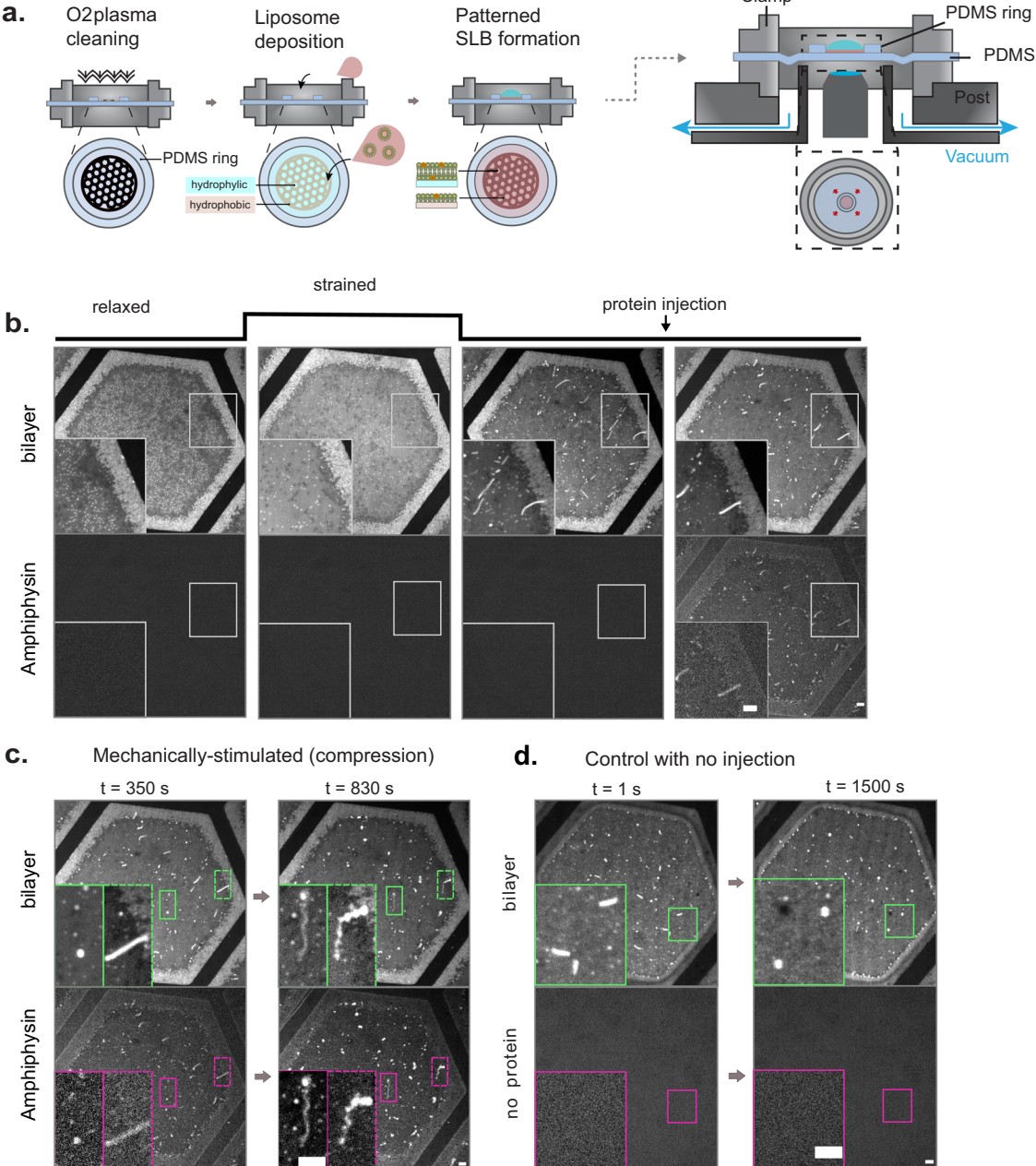

**Fig. 1 Experimental system. a** Schematics of the patterned supported lipid bilayer (pSLB) placed in a stretch system compatible with confocal microscopy. The pSLB is obtained by plasma cleaning a PDMS membrane in presence of a TEM grid. Only the exposed PDMS becomes hydrophilic, and subsequent liposome deposition renders a SLB after buffer rinse. The non-exposed PDMS remains hydrophobic and a lipid monolayer is formed instead. **b** Representative images of the mechanical stimulation of the pSLB, showing both lipid and protein fluorescence images. In the resting initial state, excess liposomes stand on top of the pSLB. With strain, the liposomes incorporate in the pSLB. Upon release, excess lipids are expelled in the form or tubes or buds. At this stage, fluorescent Amphiphysin is gently microinjected on top of pSLB and its binding to the tubes and buds is monitored with time. **c** Membrane tubes (green inset) and buds (purple inset) before (left) and after (right) being reshaped by Amphiphysin. **d** Control in which no protein is injected on top of the pSLB. Scale bar, 5 μm.

in the membrane, likely reflecting membrane tearing. (Supplementary Fig. 2e, f and Supplementary Movies 6 and 7).

To understand the physical mechanisms underlying our observations, we developed a theoretical framework considering the dynamics of lipid tubes and buds with low coverage (since protein is injected once structures are formed) and low curvature (since the structures are made markedly thinner by Amphiphysin) upon exposure to BAR proteins. Theoretically, various computational studies using coarse-grained simulations of elongated and curved objects moving on a deformable membrane have suggested the self-organization of regions with high anisotropic (cylindrical) curvature with high-protein coverage and strong nematic order[27–29]. None of these works, however, predicted or observed the tube-sphere complexes that appear in our experiments (Fig. 1c). To address this, we first focused on our recently developed mean-field density functional theory[30] for the free energy $F_{prot}$ of an ensemble of curved proteins on a membrane as a function of protein area coverage $\phi$, orientational

order as given by a nematic order parameter $S$, and membrane curvature. This theory accounts for the entropic and steric interactions between proteins and for their bending elasticity, focusing on the scaffolding effect. We discuss the mapping between different mechanisms coupling protein coverage to membrane curvature for Amphiphysin (scaffolding, effect of insertions, bulky disordered domains) and detail our model in Supplementary Note 1, where we specifically quantify the role of crowding of disordered domains by adapting a model for the coupling between such domains and curvature[31] to the present context, finding that the effect is small.

**Reshaping occurs through an isotropic-to-nematic transition**. On flat membranes and for elliptical particles of the size and aspect ratio of Amphiphysin, the theory recovers a classical entropically-controlled discontinuous isotropic-to-nematic transition during which the system abruptly changes from low to high order as protein coverage increases above $\phi \approx 0.5$, in agreement with previous results in 3D[32], Supplementary Fig. 3a1. We then examined the protein free-energy landscape on curved surfaces, where the elastic curvature energy of proteins depends on their intrinsic curvature and on the curvature of the surface along the protein long direction (Fig. 2a). On spherical surfaces and according to the theory, this energy landscape coincides with that of the flat membrane with a bias proportional to $\phi$ times the bending energy of proteins on the curved surface. Thus, the minimum energy paths as density increases (red dots in Fig. 2b–e) and hence the abrupt isotropic-to-nematic transition persists regardless of sphere radius, Fig. 2b, c, noting that on a complete sphere the nematic phase necessarily involves defects[27]. On cylindrical surfaces, however, curvature is anisotropic and the energy landscape is fundamentally modified according to our theory, as proteins can lower their free energy by orienting along a direction of favorable curvature. The competition between protein bending and entropy results in a continuous isotropic-to-nematic transition (Fig. 2d) and a significant degree of orientational order even at low coverage when the tube curvature is comparable to that of the protein (Fig. 2e). The model thus predicts how the nematic ordering of the curved and elongated membrane depends on coverage, curvature, and curvature anisotropy.

We then studied whether the model predicted the experimentally observed coexistence of thin tubes (which according to the theory should have higher coverage and order) and larger spheres (which should have lower coverage and isotropic organization). We examined the energy landscape along the minimizing paths (red dots) for spheres and tubes of varying radius (Fig. 2f). Since the slope of these curves is the chemical potential of proteins on the membrane, which tends to equilibrate with the fixed chemical potential of dissolved proteins in the medium, points of chemical coexistence are characterized by a common slope (red circles). This figure shows the largely non-unique combinations of geometry and membrane coverage compatible with coexistence in chemical equilibrium between higher-coverage nematic phases on cylinders and lower-coverage isotropic phases on spheres, supporting plausibility of such coexistence in the dynamical structures.

Shape, however, is also a dynamical variable and the selection of protein organization and shape requires the two-way interplay between the chemical-free energy $F_{prot}$ and the elastic free energy of the membrane $F_{mem}$. To account for this and for the out-of-equilibrium nature of our experiments, we self-consistently coupled a parametrization of the mean-field energy density functional theory used above with a continuum model for lipid membrane reshaping and hydrodynamics[33–35]. The combined model accounts for both energies, $F_{prot} + F_{mem}$, for the dynamics of protein adsorption from a bulk reservoir, for the diffusion of proteins on the surface, and for the membrane dissipation associated with shape changes[31] (see Supplementary Note 1, for a discussion of the model, its implementation and its parameters). Focusing on a single membrane protrusion in mechanical equilibrium (tubular or spherical, Fig. 2g) off a supported bilayer circular patch in the absence of proteins, this model predicts the dynamics of membrane shape, $\phi$, and $S$ following a sudden increase of dissolved protein concentration in the medium (Fig. 2h). In all simulations, we fixed the chemical potential of proteins in the medium to account for the dissolved protein reservoir. Once the system is driven out-of-equilibrium, we must choose a mechanical ensemble controlling the ability of the simulated system to exchange membrane area and enclosed volume with its surroundings during the dynamics. To cover the experimental conditions, we considered a reference mechanical ensemble, used everywhere unless explicitly stated, and tested the robustness of our results by further considering a broad range of ensembles with varying ease of membrane and volume exchange. For membrane exchange, we interpolated between fixed tension (allowing for membrane exchange) and fixed projected membrane area (no membrane exchange). For water exchange, we considered adhesion potentials with different strengths and a fixed pressure difference ensemble. Soft potentials allow for changes in the distance between the adhered part of the membrane and the substrate, and hence for enclosed volume exchange, as does the fixed pressure ensemble, whereas protrusion volume was nearly fixed by considering very stiff adhesion potentials (see Supplementary Note 1).

**Dynamic reshaping of buds and tubes**. We then compared model predictions with the experimental setup, by monitoring the reshaping of the buds or tubes formed after pSLB compression and upon subsequent injection of Amphiphysin at various nominal bulk concentrations. First, we examined how the mechanically-formed buds were reshaped in time as Amphiphysin binding occurs. In both our experiments and simulations, we systematically observed the growth of a thin tube emerging from the base of the bud, connecting the bud to the supported bilayer (Fig. 3a, and Supplementary Movies 8, 9, and 10). Such bud elongation from its neck also occurred upon exposure to 0.3 μM of non-fluorescently labeled Amphiphysin (Supplementary Fig. 4a and Supplementary Movie 11). According to our model, this elongation is due to a dynamic and progressive transition between two coexisting states of the membrane-protein system, one with low protein coverage, isotropic organization, and low curvature (spherical and flat parts) and another with high coverage, high nematic order, and high anisotropic curvature (thin elongating necks), Figs. 2h, 3a, and Supplementary Movie 8. The model predicts that the curvature of thin tubes is comparable to the intrinsic curvature of proteins, about 15 nm$^{-1}$, Supplementary Fig. 2b. This transition is driven by the lower bending energy of proteins at the thin neck, which outweighs both the entropic penalty of a local protein enrichment and nematic organization (Fig. 2f) and the higher membrane curvature energy of a tube relative to a larger vesicle. Our simulations also showed that protein delivery to the enriched region overwhelmingly occurs by adsorption from the bulk rather than from membrane diffusion from neighboring regions (Supplementary Note 1). Our model does not account for thermal fluctuations of shape, which have been shown to play an important role in reshaping by BAR proteins of initially planar membranes[36]. Here, reshaping is directed by membrane templates, particularly membrane necks. In this case, estimated protein coverage fluctuations resulting

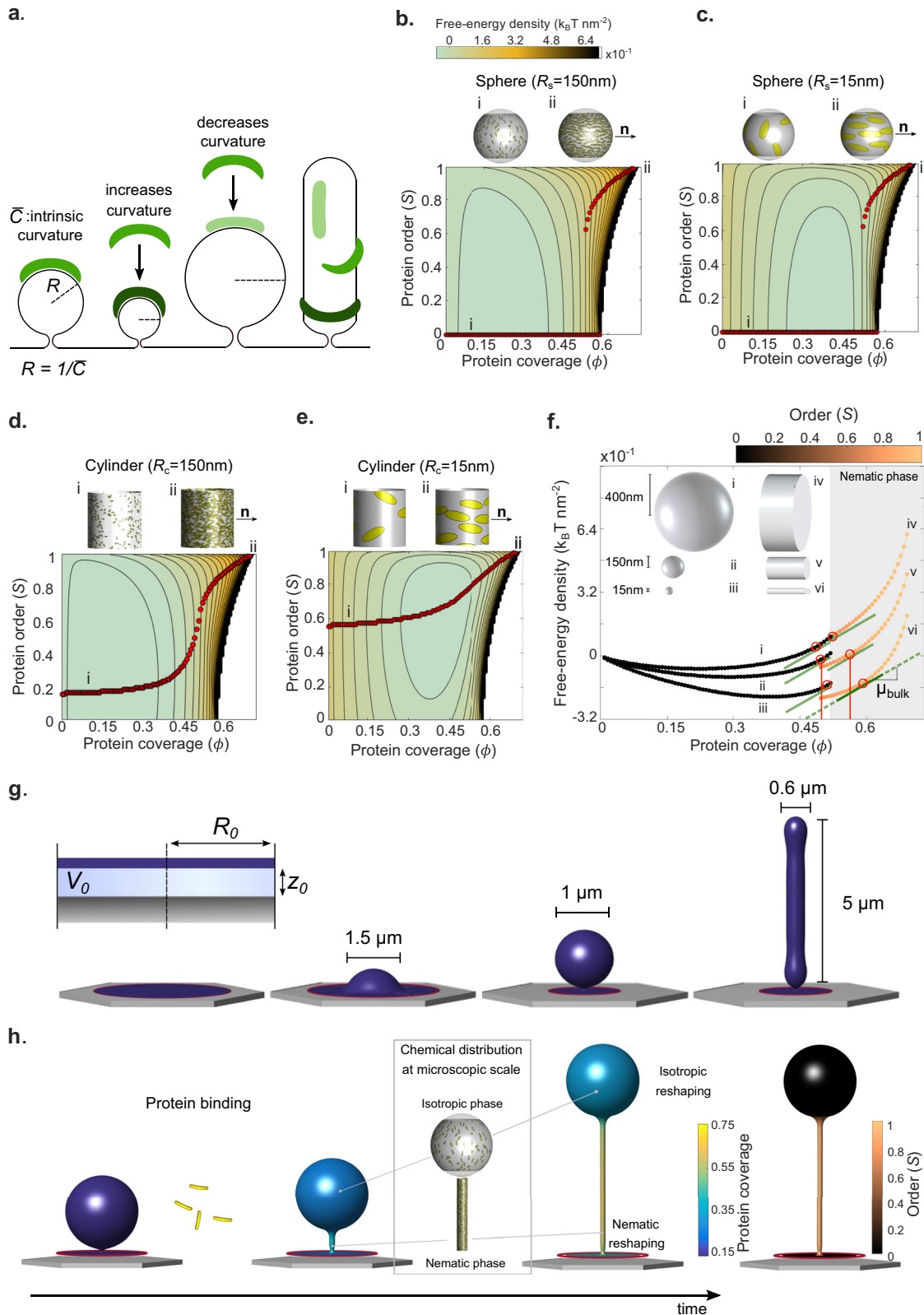

from thermal fluctuations of tubes were small (Supplementary Movie 9, Supplementary Note 1) and could be neglected.

Shape transformations in vesicles are strongly influenced by the relation between area and enclosed volume, quantified by the non-dimensional reduced volume[37]. We thus wondered about changes in membrane area and enclosed volume in our evolving protrusions. The membrane used for tube elongation may come

from the vesicle or the surrounding membrane; in experiments, we clearly observed tube elongation at the expense of vesicle area, indicated by a decrease of the vesicle diameter as the thin tube grows (as quantified experimentally in Supplementary Fig. 4b, left), until the point at which the diffraction limit does not allow to pursue reliable measurements. This suggests that membrane area is transferred from the vesicle to the elongating tube. If we

**Fig. 2 Theoretical and computational modeling. a** Schematic diagram of a BAR domain interacting with a lipid membrane. Protein elastic energy depends on surface curvature and protein orientation. For cylindrical surface, curvature is maximal (dark green) and minimal (light green) along perpendicular directions. **b–e** Landscape of free-energy density per unit area $F_{prot}$ according to our mean-field density functional theory (Section 4.1 of Supplementary Note 1) depending on protein coverage $\Phi$, nematic alignment $S$, and the shape and size of the underlying membrane (sphere or cylinder as illustrated on top of each plot, where we have generated microscopic realizations of molecular organization consistent with coverage and orientational order of the mean-field theory using a Monte Carlo algorithm). Red dots denote states of equilibrium alignments $S$ for a given protein coverage $\phi$, i.e., minimizers of the free energy along vertical profiles, depicting the transition from isotropic (i) to nematic phase (ii–iii). The white region in the energy landscape is forbidden due to steric protein interactions. **b**, **c** Discontinuous transitions for protein alignment on isotropically curved membranes. **d**, **e** Continuous transitions for anisotropically curved membrane. The intrinsic protein radius of curvature is $\frac{1}{C} = 15$ nm (see Supplementary Note 1 for other model parameters). **f** Free-energy density profiles for spheres and cylinders of different sizes along the equilibrium paths. The chemical potential of proteins is the slope of these curves. All points marked with red circles have the same chemical potential at the tangent points $\mu_b$ and hence are in chemical equilibrium. **g** Membrane protrusions obtained by lateral compression of an adhered membrane patch of radius $R_0$ interacting with a substrate with a potential $U(z)$ and for various amounts of enclosed volume $V_0$, see Supplementary Note 1. **h** Schematic of reshaping dynamics involving membrane relaxation, and protein binding, diffusion, and ordering.

compare a bud with an elongated tube with the same area, the tube would have a much lower enclosed volume. This suggests that this transformation requires significant enclosed volume exchange, explaining why simulations with our reference ensemble did not lead to significant vesicle shrinking during tube elongation, Fig. 3a and Supplementary Movie 8, but simulations with an ensemble fixing the membrane area of the protrusion and enabling easy water exchange replicated this aspect of the membrane reshaping (Supplementary Fig. 4b, right, Supplementary Movie 12).

Then, we considered the reshaping dynamics of tubes, which were frequently formed upon compression. In this case, we systematically found both in experiments and simulations that tube reshaping was initiated by the formation of a sequence of pearls (Fig. 3b and Supplementary Movies 13–15). Tube pearling also occurred upon exposure to 0.3 μM of non-fluorescently labeled Amphiphysin (Supplementary Fig. 4c and Supplementary Movie 16). Subsequently, the pearled tubes transformed into pearls connected by thin tubes. This configuration was stable for long time (though such reshaped tubes collapsed on themselves, likely due to the related loss of tension[3,38]—this phase is best observed in the movie, see Supplementary Movie 14). Necks progressively elongated, eventually making pearls disappear and transforming the structure into a long thin tube.

To understand this process, we noticed that in our simulations pearling occurred at low and nearly uniform protein coverage with nearly isotropic organization, Figs. 2d, 3b, and hence this transformation can be ascribed to a previously described pearling instability in the presence of sufficiently large isotropic and uniform spontaneous curvature[31,39,40]. Indeed, pearling in simulations occurred even without nematic order, Supplementary Fig. 4d. Although this first step is independent of molecular alignment, it triggers nematic order, as pearling generates several thin necks along the tube. These necks nucleate regions of high anisotropic curvature, high coverage and nematic order coexisting with low-curvature spheres with low-coverage and isotropic molecular arrangement and subsequent tube elongation between the pearls. If the protein chemical potential is high enough, the necks progressively elongate into tubes while the spheres progressively disappear. We witness here a striking process in which a large tube (compared with the N-BAR dimer intrinsic curvature) is reshaped by Amphiphysin at low coverage due to an isotropic rearrangement of the proteins, giving rise to nucleation points of thin tubular necks promoting further protein enrichment and nematic ordering.

**Physical parameters governing membrane reshaping**. We then addressed the time-scale of the reshaping process. For vesicles exposed to Amphiphysin, we measured tube elongation rates

between 20 and 75 nm/s at 0.25 and 0.35 μM nominal concentrations, and from 365 to 550 nm/s at 0.5 μM. These rates were much smaller, about two orders of magnitude, than those predicted by our simulations, Supplementary Fig. 4b. To address these large differences in time-scale between simulations and experiments, we turned to the dynamics of the injection process, not accounted for in our model. Since our protein injection method was systematic (Supplementary Fig. 5a), we modeled protein delivery through bulk diffusion in the medium from the injection point to the close vicinity of the SLB. We estimated the time evolution of protein concentration to which the SLB is effectively exposed by solving a diffusion equation, readily providing a mapping between time and concentration at the SLB depending on the bulk nominal concentration (Supplementary Figs. 5a and 6a). According to this analysis, diffusion in the bulk is the slowest process as compared to adsorption, membrane diffusion, and membrane mechanical reshaping (see Supplementary Note 1), and thus it controls reshaping and explains quantitatively the very different times at which a given state is observed for different nominal concentrations (Supplementary Figs. 5c and 6c). It also explains that in our experiment, the higher the bulk nominal concentration, the faster reshaping occurred (for buds, Supplementary Fig. 5b, c and Supplementary Movies 9 and 10, for tubes, see Supplementary Fig. 6b, c and Supplementary Movies 13 and 14). When we performed simulations as a succession of quasi-equilibrium states at increasing concentrations up to the nominal concentration (Supplementary Figs. 5d and 6d), we found the same reshaping mechanisms as in our dynamical simulations of buds and tubes where the nominal concentration was applied instantaneously (Fig. 3a, b). More specifically, for buds, both approaches exhibited the nucleation and elongation of a tubular nematic and protein-rich phase at the bud neck, and for thick tubes they both proceeded by a pearling instability at low-coverage low-order protein states followed by nucleation and growth of tubular nematic and protein-rich phases at the necks between the pearls. Hence, we can interpret our experimental observations as quasi-equilibrium states at a given dissolved protein chemical potential in the close vicinity of the SLB.

We further tested the robustness of the reshaping mechanisms identified above by varying the mechanical ensemble governing exchange of membrane area and enclosed fluid volume in protrusions as they deform (see Supplementary Note 1). Supplementary Fig. 7 displays snapshots of the shape of the buds or tubes at their onset of reshaping or after further elongation at similar bulk concentrations to allow for comparison and for a selection of the tested ensembles. We found that, although the thresholds for reshaping and the reshaping progression at a given bulk concentration slightly depended on

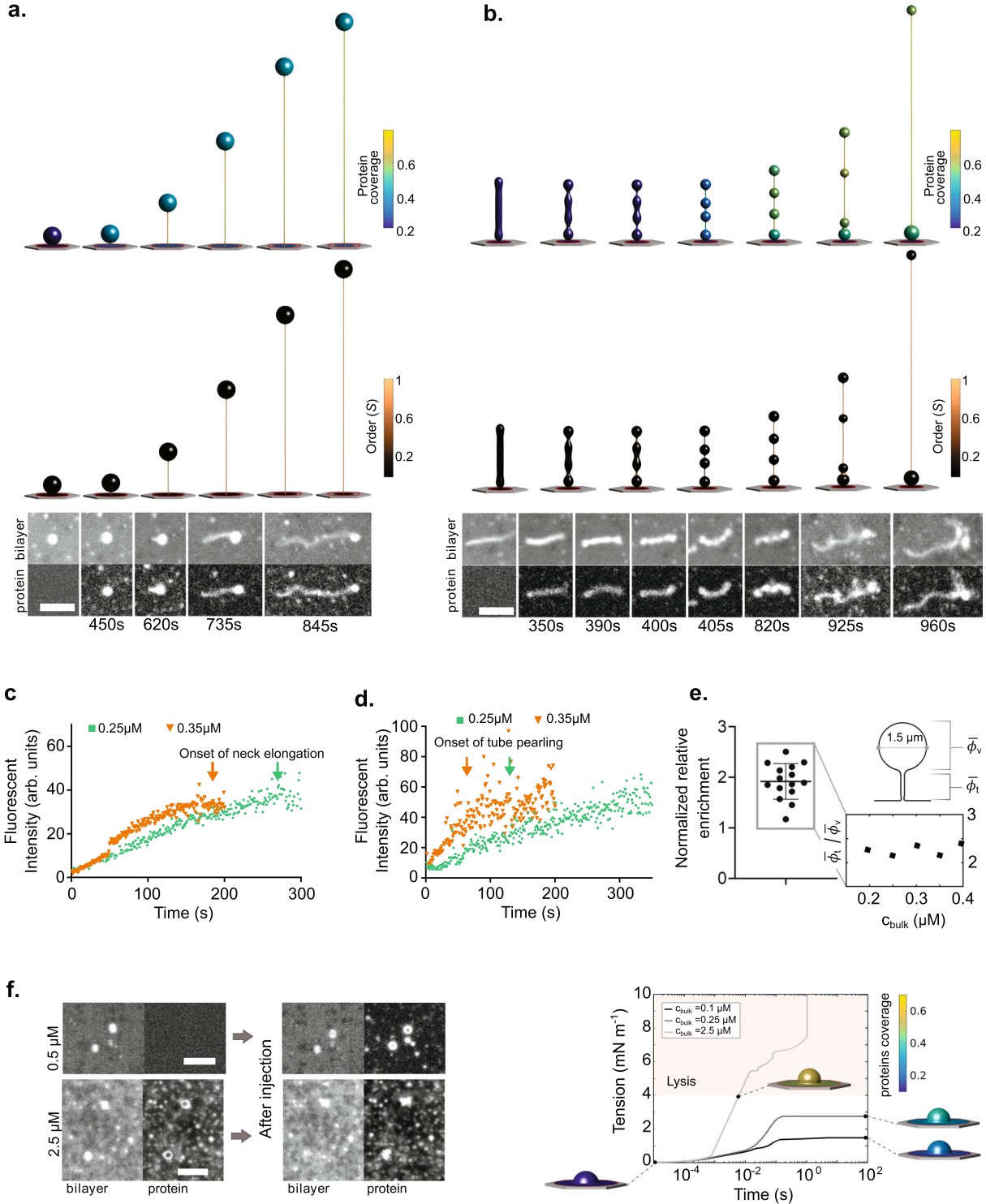

**Fig. 3 Dynamics of membrane reshaping. a** Results of simulations (top) and experiments (bottom) showing bud reshaping with time in response to Amphiphysin. Scale bar, 5 μm. **b** Results of simulations (up) and experiments (down) showing tube reshaping with time in response to Amphiphysin. These simulations were performed using the reference mechanical ensemble described in Supplementary Note 1. Scale bar, 5 μm. **c** Examples of Amphiphysin fluorescence intensities in buds incubated at two different concentrations. Bud elongation times are marked with an arrow. Source data are provided as a Source data file. **d** Examples of Amphiphysin fluorescence intensities in tubes incubated at two different concentrations. Tube pearling times are marked with an arrow. Source data are provided as a Source data file. **e** Ratios of protein coverage on tubes versus buds, normalized to the values measured for the lipid bilayer. Left: experimental values ($n = 15$), right: theoretical concentration ratios $\frac{\phi_t}{\phi_v}$ for a 1 μm diameter bud, exposed to different bulk concentrations. Data are shown as mean ± s.d. and source data are provided as a Source data file. **f** Left: Initial and final states of pressurized caps (obtained from an hypoosmotic shock) upon incubation with Amphiphysin. At 2.5 μM concentration, lysis of the caps can be observed. Scale bars, 5 μm. Right: Model prediction in pressurized caps of about 1.5 μm in diameter in radius exposed to different Amphiphysin concentrations. States in the pink shaded area are prone to membrane lysis.

the mechanical constraints, the fundamental mechanisms of reshaping regions were consistent irrespective of the mechanical ensemble. Those consisted in nucleation (either at pre-existing necks for buds or at necks generated by the pearling transition in tubes) and in the elongation of highly curved tubes with high-coverage and nematic order, coexisting with low-curvature, low-coverage, and isotropically organized regions. The only exception, not observed in our experiments, was the reshaping of tubes in the specific condition of no membrane exchange and very easy volume exchange by fixing pressure across the membrane, in which case the high-curvature high-coverage nematic tubular state was reached by progressive elongation and thinning, Supplementary Fig. 8.

We also varied the size of the protrusions to capture the heterogeneity in sizes and shapes experimentally obtained, with bud diameters ranging from 0.5 to 1.5 μm, and tube lengths from 2 to 5 μm, and a diameter of 600 nm as discussed in Supplementary Note 1, section 5. We consistently found that the fundamental features of the previously described reshaping process were independent of protrusion dimensions (Supplementary Fig. 7, Supplementary Movies 17 and 18).

Finally, as the model predicts coverage as the key parameter that controls reshaping, we explored the coverage at which the onset of tubulation for buds, or pearling for tubes, occurred. Experimentally, we plotted the protein binding curves on the buds or on the tubes obtained at different bulk nominal concentrations, by displaying the mean intensity of the protein fluorescence on buds over time (Fig. 3c, d). The onset of reshaping occurred faster at higher concentration, however, they started at a comparable intensity level regardless of the nominal bulk concentration. Then, we developed a protocol to obtain an estimate of protein coverage from fluorescence levels. We performed a classical calibration of the protein fluorescence versus coverage[41], and included a geometrical correction, by taking as a reference the fluorescence from the lipid channel (see Supplementary Fig. 9a). This enabled us to correct for the increase of fluorescence due to integration of a fluorescence from a 3D structure when taking 2D images. It also corrects for the loss of signal since we are not exactly focusing on the bilayer plane, but a bit above to resolve better the 3D geometry of the moving templates. We fitted the data with an exponential curve, and averaged the coverage values obtained at bud elongation and tube pearling. As a result, initiation of bud elongation or tube pearling occurred at respectively 0.44 +/− 0.097 and ~0.34 +/− 0.08 coverage (Supplementary Fig. 9b). Then, we analyzed the corresponding theoretical predictions for the onset of reshaping as a function of the mechanical ensemble and size, Supplementary Fig. 9c. Theoretical predictions mildly depended on the mechanical ensemble and size, but generally matched experimental results well for bud elongation. In the case of tube pearling they were slightly lower, likely due to the experimental difficulty of precisely capturing the onset of pearling (in contrast to bud elongation, which is a much more obvious event). Besides coverage, another hallmark of this isotropic-nematic coexistence is a protein enrichment on the tube relative to the vesicle. We estimated this experimentally (Supplementary Fig. 10a) and measured approximately a two-fold higher protein concentration in tubular versus bud regions (Fig. 3e). For a wide range of bud diameters and protein concentrations, our simulations predicted a comparable enrichment (Fig. 3e and Supplementary Fig. 10b). Our finding is in good agreement with a related study[4] where enrichment ranging from 1.8 to 5 were reported on tubes pulled from a giant vesicle.

Taken together, our results show the complete path through which low curvature spherical or tubular templates exposed to BAR proteins evolve toward uniformly thin and protein-rich

nematic tubes. This process involves a non-homogeneous, mechanochemical transition involving a low-curvature, low-coverage spherical state with isotropic molecular (phase I) organization and a high-curvature, high-coverage nematic tubular state (phase II). Indeed, at low to moderate protein coverages, heterogeneous intermediates are formed, exhibiting mixtures phases I and II. In fact, phase II nucleates in a phase I matrix, and propagates the phase boundaries until reaching a homogeneous phase II at full protein coverage. This occurs in a dynamic process where curvature sensing and generation are integrated within the same framework.

**Reshaping is hindered by high tension**. As an additional case, we explored the behavior of the system when shape changes are not allowed. To this end, we generated shallow spherical cap protrusions, which develop when hypo-osmotic shocks are generated both in vitro and in cells[6,7]. In this case, the cap templates adopt a spherical shape and are pressurized, unlike the buds previously obtained by bilayer compression. Therefore, the membrane needs to accommodate a significant excess volume of liquid with little excess membrane area, leading to a structure under significant tension[6] where shape changes are very difficult. Accordingly, shallow spherical caps formed by a hypo-osmotic shock in our experimental system were not visibly reshaped by Amphiphysin even at significant concentrations (Fig. 3f and Supplementary Movie 19). Moreover, the cap teared and collapsed upon exposure to higher Amphiphysin concentrations (Fig. 3f and Supplementary Movie 20). Our model predicts that upon exposure of such shallow caps to BAR proteins, shape changes are negligible. The model does not explicitly describe tearing but predicts membrane tension. As protein concentration increases, tension in the membrane sharply increases, potentially leading to membrane tearing[21,42] (Fig. 3f).

**Cell compression triggers membrane tubulation by Amphiphysin**. Beyond the specifics of the reshaping process, an important conclusion from this study is that the mechanical generation of membrane structures acts as a catalyzer of membrane reshaping by BAR-domain proteins. Indeed, compressed membranes exhibited a wide range of reshaping behaviors (Fig. 3), whereas non-mechanically stimulated membranes exposed to the same Amphiphysin concentration did not reshape in any clear way (Supplementary Fig. 2e and Supplementary Movie 6). This suggests the interesting possibility that cells could harness the mechanically-induced formation of membrane invaginations[7] to trigger BAR-mediated responses, thereby enabling mechanosensing mechanisms. To explore this possibility, we cultured dermal fibroblasts (DF) and overexpressed GFP-Amphiphysin, which is well known to trigger spontaneous membrane tubulation[20]. Then, we stretched and subsequently compressed the cells using a previously described protocol[7]. Upon compression, cells formed dot-like membrane folds termed "reservoirs" (Fig. 4a), analogous to the membrane structures observed in vitro in Figs. 1 and 3. Amphiphysin-containing membrane tubes formed before, during, and after stretch. However, their number decreased during the stretch phase, likely due to increased membrane tension (Fig. 4c). Upon release of the stretch, tube formation strongly increased, reaching values well above the initial non-stretched condition (Fig. 4c and Supplementary Movie 21). Further, tubes formed upon de-stretch nucleated close to reservoir locations (Fig. 4b). We measured the elongation rates of these tubes and they ranged from 200 to 350 nm/s, comparable with the elongation rates found in the pSLB experiments for bud elongation. Though Amphiphysin overexpression presumably leads to concentrations above

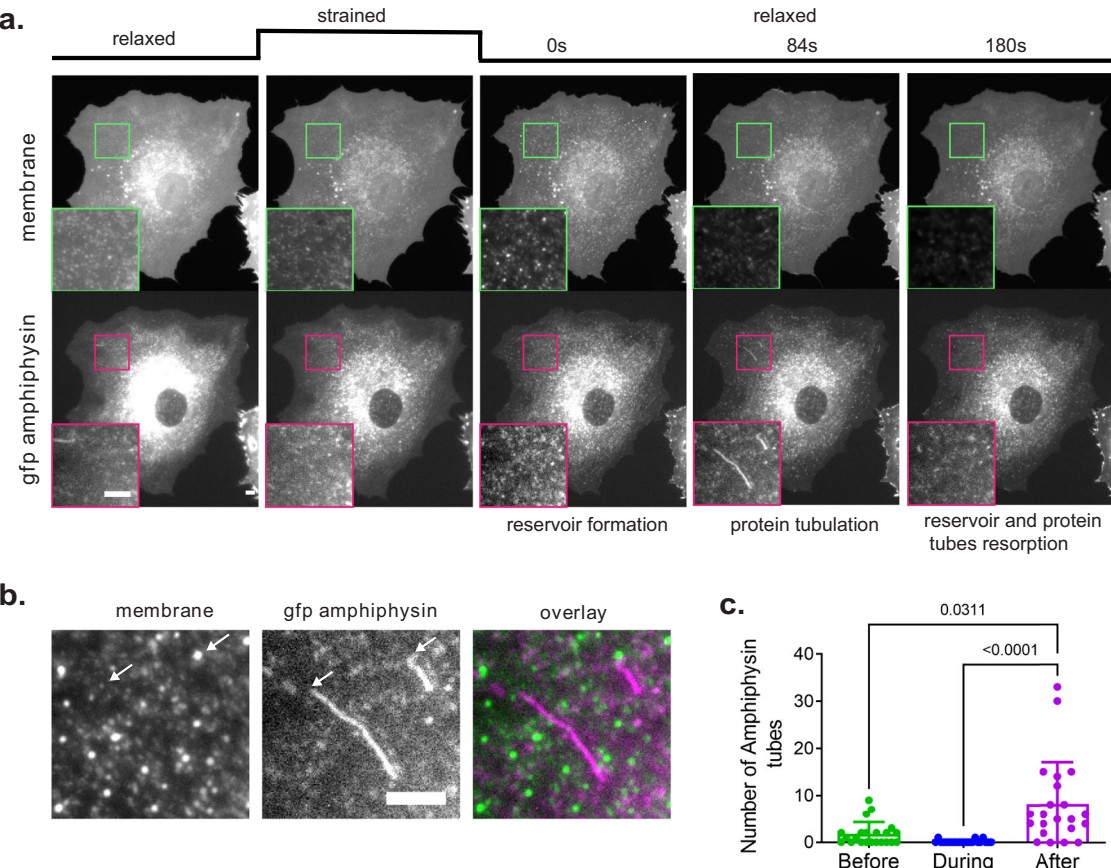

**Fig. 4 Mechanical stretch in cells triggers Amphiphysin-mediated tubulation. a** Representative images (in both membrane and Amphiphysin channels) of a cell before, during, and after stretch release. **b** Detail of membrane and Amphiphysin channels during tubulation. **c** Quantification of the number of Amphiphysin tubes at rest, during stretch, and once stretch is released ($n = 22$, Friedman test (two-tailed)). Data are shown as mean ± s.d., and source data are provided as a Source data file. Scale bars, 5 µm.

physiological levels, these results clearly show that mechanical compression of cells can stimulate not only BAR-protein recruitment (and possible ensuing signaling cascades) but also BAR-mediated membrane tubulation, abruptly affecting plasma membrane shape.

## Discussion

Curvature sensing and membrane reshaping properties of BAR proteins have been extensively studied on highly curved tubes (up to 100 nm in diameter) mostly in equilibrium[4,43–45], but the dynamics of the process and its dependence on the initial template were unexplored. In this work, we present a charged synthetic lipid bilayer system that stores sufficient bilayer area and allows for significant bilayer stretch. With stretch release, lipids accommodated in curved structures generate templates for bilayer reshaping upon Amphiphysin binding. Our system is complementary to other in vitro systems, that typically consider either tubes pulled under tension,[3,4] or free-standing liposomes or tubes uncoupled from any external lipid reservoir[46,47]. Instead, our system generates a heterogeneous population of shapes at low tension, curvature, and concentration, a highly relevant scenario in cells, and evaluates BAR protein reshaping dynamically. Our accompanying model provides a mechanistic explanation of this dynamic reshaping, which conforms a very rich process with many intermediate steps, including non-homogeneous phase separation between isotropic and nematic phases, and with major reshaping processes occurring at low coverage and curvature. This behavior emerges naturally from the fundamental physics of

membrane mechanics and its mechanochemical interactions with curved proteins, generating a non-trivial feedback between membrane mechanical stimulation and subsequent response. Beyond the physics of the process, such feedback could potentially be used in the many cellular processes involving membrane reshaping. Indeed, the physiological role of BAR proteins at low concentration is mostly studied in the context of BAR protein sensing of highly curved structures, but many cell studies have shown BAR proteins acting on lower curvature structures[48], where reshaping is expected to occur out of equilibrium. This study provides a mechanistic framework to understand how BAR protein remodeling in such a context may occur. This may be relevant in well-studied processes such as endocytosis, but also in emerging roles of BAR proteins in maintenance of cell polarity[14], response to osmotic changes[49], or build-up of caveolar structures[12,13]. Our results also open the door to unexplored scenarios involving reshaping of low tension, low-curvature membranes obtained under mechanical constraints.

## Methods

**Protein expression and purification**. The plasmid containing full-length human Amphiphysin 1 (FL-hAMPH), pGEX-Amphiphysin1, was a kind gift from Pr. De Camilli, Yale University. The plasmid codes for the FL-hAMPH preceded by a Glutathione S-Transferase (GST-Tag) and a cleavage site recognized by prescission protein. The plasmid was transformed in *Escherichia coli* Rosetta™ (DE3) pLysS cells (Novagen). Selected colonies were grown in luria broth supplemented with 25 µg/ml chloramphenicol and 25 µg/ml kanamycin at 37 °C until an OD between 0.6 and 0.8 was reached. Protein expression was induced by 1 mM isopropyl-β-D-thiogalactopyranosid (IPTG) overnight at 25 °C. Cells were pelleted for 30 min at 3,315 × *g*, pellet was resuspended in lysis buffer (10 mM phosphate buffer saline pH 7.3, supplemented with cOmplete protease inihibitor, EDTA free (Roche) and

1 mM Phenylmethylsulfonyl fluoride (Sigma)). Cells were lysed (5 pulses of 30 s sonication with 30 s rest), incubated for 20 min on ice with 5 µg/ml DNase, and centrifuged at $75,000 \times g$ for 45 min. The supernatant was collected and incubated with 2 column volume (for 20 mL supernatant) of Gluthatione Sepharose 4B (GE Healthcare) for 1 h 30 min on a rotating wheel. The beads were subsequently washed with phosphate buffer saline buffer, pH 7.3 before exchanging to the cleavage buffer (50 mM Tris-base, 150 mM NaCl, 1 mM EDTA, 1 mM DTT, pH 7.0). Sixty units of Prescision protease (BioRad Laboratories) were added to the beads and cleavage of the GST-Tag was allowed for 1 h at room temperature followed by an overnight incubation at 4 °C on a rotating wheel. The flow through was recovered, and contained cleaved Amphiphysin that was further purified by size exclusion chromatography in a Superdex 75 26/60 in 10 mM PBS, pH 7.5, 1 mM DTT. Two fractions were obtained, both containing Amphiphysin according to the SDS-page gel, but the second fraction of smaller size was taken and concentrated for further use. The purity and identity of the product was established by HPLC and mass spectrometry (BioSuite pPhenyl 1000RPC 2.0 × 75 mm coupled to a LCT-Premier Waters from GE Healthcare). Neutravidin was from Thermofisher. Proteins (Amphiphysin and Neutravidin) were coupled to an Alexa Fluor® 488 TFP ester according to the manufacturer protocol and the resulting protein-alexa 488 was concentrated again. Adsorption was measured in a Nanodrop at 280 nm to obtain protein concentration and at 488 nm to obtain fluorophore concentration. This gave an average amount of fluorophore per protein of 3 per Amphiphysin dimers and 1 per Neutravidin protein. Amphiphysin was frozen and kept at −80 °C, experiments were performed with freshly unfrozen samples. Protein integrity was verified by SDS-page of the unfrozen samples.

**Preparation of stretchable membranes**. Stretchable polydimethylsiloxane (Sylgard Silicone Elastomer Kit, Dow Corning) membranes were prepared as previously described[7]. Briefly, a mix of 10:1 base to crosslinker ratio was spun for 1 min at 500 rpm and cured at 65 °C overnight on plastic supports. Once polymerized, membranes were peeled off and assembled onto a metal ring that can subsequently be assembled in the stretch device.

**Patterned supported lipid bilayer (pSLB) formation on PDMS membrane**. pSLBs were prepared by combining 1,2-dioleoyl-sn-glycero-3-phosphocoline (DOPC), 1,2-dioleoyl-snglycero-3-phospho(1'-rac-glycerol) (sodium salt) (DOPS), and 1,2-dioleoyl-sn-glycero-3-phosphate (sodium salt) (DOPA), 1,2-dipalmitoyl-sn-glycero-3-phosphoethanolamine-N-(lissamine rhodamine B sulfonyl) (ammonium salt) (LissRhod-DPPE). 1.25 mg of total lipids in a DOPC:DOPS:DOPA 3:2:1 proportion, with 0.5% mol LissRhod-DPPE were dissolved in chloroform. Addition of 1,2-dioleoyl-sn-3-phosphoethanolamine (DOPE) as an alternative to DOPA in the pSLB did not allow for a fluid bilayer formation. For control experiments, biotinylated pSLBs were prepared with 1.25 mg DOPC with 0.5% mol LissRhod-DPPE and 5% mol 1,2-dipalmitoyl-sn-glycero-3-phosphoethanolamine-N-(cap biotinyl) (sodium salt) (16:0 Biotinyl Cap PE). We consistently used the same acyl chains in the lipid composition in order to minimize asymmetry between the two leaflets. The solvent was evaporated for a minimum of 4 h. The lipid film was immediately hydrated with 750 µL of PBS, pH 7.5 (final concentration of 1.6 mg/ml) at room temperature. After gentle vortexing, a solution of giant multilamellar vesicles was obtained. Large unilamellar vesicles (LUVs) were prepared by mechanical extrusion using the Avanti extruder set. The lipid suspension was extruded repeatedly (15 times) through a polycarbonate membrane (Whatman® Nuclepore™ Track-Etched Membranes diam. 19 mm, pore size 0.05 µm). The mean diameter of the LUVs was verified by Dynamic Light Scattering (Zetasizer Nano-series S, Malvern instruments). LUVs were always prepared freshly the previous day of the experiment.

To prepare the pSLB, a TEM grid (G200H-Cu, Aname) was placed in the middle of the PDMS membrane ring. The membrane was subsequently plasma cleaned in a Harrick oxygen plasma cleaner using the following parameter: constant flow of oxygen between 0.4 and 0.6 mbar, high power, and exposure time between 15 and 60 s. A small 6 mm inner diameter ring was simultaneously plasma cleaned and bonded around the TEM grid. Then, the TEM grid was removed and the liposome solution was deposited and confined inside the thin bonded ring, with subsequent incubation for 1 h at room temperature. LUVs were then extensively washed with PBS buffer pH 7.5. The membrane was mounted in the stretching device placed in the microscope.

**FRAP of the pSLB**. Patterned Supported Lipid Bilayers (pSLB) were obtained as described above on PDMS membranes, and the ring-containing membranes were mounted under an upright epifluorescence microscope (Nikon Ni, with Hamamatsu Orca Flash 4.0, v2). Images of pSLBs, obtained with either 15 or 30 s plasma cleaning, were acquired with a 60x water dipping objective (NIR Apo 60X/WD 2.8, Nikon) and an Orca R2 camera. A small linear region of the pSLB was frapped by repeatedly scanning and focusing 180 fs pulses generated by a fiber laser (Femto-Power, Fianium) with central wavelength at 1064 nm at 20 MHz. A set of galvo mirrors (Thorlabs) and a telescope before the port of the microscope allowed to position and move (oscillations at 400 Hz) the diffraction-limited spot at a desired place on the bilayer. Once bleached, fluorescence recovery was monitored for 5 min. Time-lapse imaging during the pSLB photobleaching and its recovery after

photobleaching was done with a home-made software (Labview 2011). Recovery of the intensity of the bleached lines were plotted either for the full line, or by separating the line into a left and a right area and assess whether the recovery was symmetric (Supplementary Fig. 1).

**Sucrose-loaded assay and negative-stain transmission electron microscopy**. Sucrose-loaded vesicles were prepared as previously described in the literature[24], using a mixture of DOPC, DOPS, and DOPE lipids in a 1:2:1 ratio. Lipids were evaporated and subsequently rehydrated with PBS buffer pH 7.5, containing 0.3 M sucrose. A solution of 0.6 mM lipids of vesicles was incubated for 20 min with 40 µM of Amphiphysin (non-fluorescent) at 37 °C. The solution was incubated on a copper grid (G200H-Cu + Formvar, Aname), previously activated (with 5 min UV) and subsequently stained with 2% neutral phosphotungstic acid. Grids were imaged in a JEOL 1010 80 kV TEM microscope, and recorded with the AnalySIS software.

**Mechanical/osmotic stimulation of the pSLB, protein injection, and live imaging**. Membrane-containing rings were mounted in the stretch system as previously described[7]. Image acquisition of cells and pSLBs were acquired with a 60x objective (NIR Apo 60X/WD 2.8, Nikon) in an inverted microscope (Nikon Eclipse Ti) with a spinning disk confocal unit (CSU-W1, Yokogawa), a Zyla sCMOS camera (Andor) and using the Micromanager software. The bilayer was stretched slowly for 120 s and the strain, obtained through the measurement of the hexagon extension, was between 5 and 8%. After 120 s stretch, the bilayer was slowly released for 300 s. At release and upon tube appearance, images were acquired every sec in two different channels collecting each fluorophore emission signal. Given the 3D structure of the tubes and buds, manual focusing enabled to image these lipid templates over time, slightly above the bilayer plane. Three microliters of an Amphiphysin or Neutravidin stock solution (of a concentration depending on the desired end concentration but always in the same buffer as the one covering the pSLB to avoid any osmotic perturbation) was gently microinjected in the buffer droplet hydrating the pSLB. End concentration ranged from 50 nM to 5 µM. In some instances, the non-fluorescent protein was used to reach high concentrations. For the controls of tube behavior in absence of protein, no injection was performed. To modify osmolarity, the pSLB was exposed to medium mixed with de-ionized water and after pressurized cap formation, protein was injected in the same conditions as above. Osmolarity was adjusted to that of the buffer hydrating the pSLB.

**Supported lipid bilayer (SLB) formation on glass coverslips**. SLBs on glass coverslips used for the calibration in the quantitative fluorescence microscopy were obtained as previously described[50]. Glass coverslips were cleaned by immersion in 5:1:1 solution of $H_2O:NH4:H_2O_2$ at 65 °C for 20 min and were dried under a stream of $N_2$ gas. GMVs were obtained as previously but with different lipid mixtures. To obtain SLBs with 0.1 to 0.5% of protein-like fluorophores, two LUV-stock solutions were prepared, either DOPC only, or DOPC with 0.5% 1,2-dioleoyl-sn-glycero-3-phosphoethanolamine-N-(TopFluor® AF488) (ammonium salt). Lipid films were rehydrated in 150 mM NaCl and 10 mM Tris, pH 7.4, to a final concentration of 3 mg/mL. GMVs were extruded as previously described to obtain LUVs. Small rings of 6 mm diameter of PDMS were bonded as described before using plasma cleaning of both substrates, forming a small chamber on top of the coverslip. Coverslips were activated by cleaning with oxygen plasma (Harrick) in a constant flow mode (pressure 0.6 and at high power) for 20 min. The two LUV-stock solutions were diluted in fusion buffer (300 mM NaCl, 10 mM Tris, 10 mM $MgCl_2$) to 0.5 mg/mL solutions at different ratios to obtain a set of solutions from 0 to 0.5% TopFluor-AF488. SLBs of the different fluorophore ratios were obtained by incubating the diluted solutions in the glass coverslips chambers, immediately after the plasma cleaning process, for 1 h at room temperature. Liposomes were extensively rinsed with the fusion buffer and subsequently milli-Q water.

**Imaging of the SLBs, liposome, and protein solutions on glass for quantitative fluorescence microscopy**. SLBs on glass were imaged in the same condition as the pSLB on PDMS. For the AF-488 enriched SLB, the exposure time and laser power were the same as for the protein channel. For the LissRhod-DPPE enriched SLB, parameters were the same as for the lipid channel. Background for the AF-488 enriched SLB was obtained by focusing on a LissRhod-DPPE enriched bilayer and recording an image in the 488 nm channel. The opposite was done for LissRhod-DPPE enriched background. Fluorescence image of protein solutions at different concentrations, from 0 to 0.75 µM, and of LissRhod-DPPE enriched LUV solutions (from 0 to 0.1%) were recorded with the same settings as for the pSLB protein channel.

**Cell culture and transfection**. Normal Human Dermal Fibroblasts derived from an adult donor (NHDF-Ad, Lonza, CC-2511) were cultured using Dulbecco's modified Eagle medium (DMEM, Thermofisher Scientific, 41965-039) supplemented with 10% FBS (Thermofisher Scientific, 10270-106), 1% Insulin-Transferrin-Selenium (Thermofisher Scientific, 41400045) and 1% penicillin-streptomycin (Thermofischer Scientific, 10378-016). Cell cultures were routinely checked for mycoplasma. $CO_2$-independent media was prepared by using $CO_2$-

independent DMEM (Thermofischer Scientific, 18045-054) supplemented with 10% FBS, 1% penicillin-streptomycin, 1.5% HEPES 1 M, and 2% L-Glutamine (Thermofischer Scientific, 25030-024). One day before experiments, cells were co-transfected with the membrane-targeting plasmid peGFP-mem and the Amph1-pmCherryN1. Transfection was performed using the Neon transfection device according to the manufacturer's instructions (Invitrogen). peGFP-mem was a kind gift from Pr. F. Tebar. and contained the N-terminal amino acids of GAP-43[51], which has a signal for post-translational palmitoylation of cysteines 3 and 4 that targets fusion protein to cellular membrane, coupled to a monomeric eGFP fluorescent protein. Amph1-pmCherryN1 was a kind gift of Pr. De Camilli and contained the full-length Amphiphysin 1 coupled to a mCherry fluorophore.

**Mechanical stimulation of the cells and live imaging**. Cell mechanical stimulation was done as previously described[7]. Briefly, a 150 μL droplet of a 10 μg/mL fibronectin solution (Sigma) was deposited in the center of the membrane mounted in the ring. After overnight incubation at 4 °C, the fibronectin solution was rinsed, cells were seeded on the fibronectin-coated membranes and allowed to attach during 30–90 min. Then ring-containing membranes were mounted in the stretch system previously described[7]. Cell images were acquired with a 60x water dipping objective (NIR Apo 60X/WD 2.8, Nikon) and an Orca Flash 4.0 camera (Hamamatsu), in an upright epifluorescence microscope with the Metamorph software. Cells were always imaged in two different channels collecting each fluorophore emission signal, every 3 s. They were imaged for 2 min at rest, 3 min in the 6% stretched state (nominal stretch of the PDMS substrate), and 3 min during the release of the stretch.

#### Quantifications

*Diameter of tubes expelled by Amphiphysin from sucrose-loaded vesicles using TEM images*. The diameter of the lipid tube reshaped by amphiphysin was measured using the TEM images from the sucrose-loaded assay. Diameters at one or two places of tubes expelled from the vesicles were measured manually on seven different high magnification images (*60k) of two independent experiments. The mean diameter was computed from these measurements.

*Binding curves of the protein to the buds and tubes*. Stacks of the acquired images were prepared in Fiji. A stack containing a single lipid object (tube or bud) was isolated from the timelapse stacks obtained in protein channel, as well as a stack of a small area of the pSLB close to the object. Objects were automatically thresholded in Cell Profiler and their mean florescence intensity was extracted. After background correction, the fluorescence intensity was plotted over time for each object.

*Protein enrichment on the reshaped tube*. The raw intensities of the elongated tubes were measured as explained above (in the tube diameter section), for both lipid and protein channels, at the same timepoint. The raw intensity of the bud in both channels was also measured assuming a spherical shape. We define the tube versus bud enrichment in both channels by the ratio between the mean intensities of the tube and bud. Mean intensities are calculated by dividing the raw intensities by the area of the tube or bud, which is the same in both channels. In the case of the lipid image, no enrichment is assumed. We thus normalize the protein enrichment value with that of the lipid which makes our measurement independent of geometry. See also Supplementary Fig. 9a.

*Estimation of protein coverage*. To estimate the coverage of tubes and buds with Amphiphysin, we first prepared flat membrane bilayers containing 0.5% Liss Rhodamine fluorophore, and measured their average fluorescence intensity per unit area. Then, tubes or buds in experiments were identified as described in the "binding curve method", and their average fluorescence intensity in the lipid channel was also calculated. By calculating a ratio between both values, we obtain a geometrical correction factor. Due to the 3D shapes of tubes and buds, this accounts for loss of signal if not all fluorescence is collected in the confocal slice, or gain of signal due to integration of fluorescence due to the 3D object.

Then, we prepared flat membrane bilayers, but labeled with the same fluorophore used for Amphiphysin, AF-488. By measuring fluorescence intensities as a function of AF-488 concentration, we obtained a calibration curve between the fluorescence signal and fluorophore concentration, as previously described[41,50]. Finally, we measured the average fluorescence intensity of tubes and buds in the Amphiphysin channel, and used the calibration curve and the geometrical correction factor to estimate an Amphiphysin dimer concentration (accounting for the number of fluorophores per dimer). After assuming a dimer area of 58 nm$^2$ (same area as in our simulations, close to the one classically used[4]), we finally obtain a coverage estimation (see also Supplementary Fig. 7a) of the protein on the tube or bud at each timepoint. We then fit the data with an exponential curve using the equation: Coverage = $C_{max}(1 - e^{-kt})$ to the experimental evolution with time of each analyzed structure, and take the coverage value at which reshaping begins. Then, we calculated the mean and standard deviation of all points.

*Quantification of Amphiphysin tubulation in the cell experiments*. In movies of Amphiphysin over-expressing cells, time slots of 90 s before, during, and after stretch were analyzed. The number of tubulations appearing in each one of the slots was manually counted having as reference the timepoint of formation of the structure. The graph and statistics were generated using the Graphpad prism software.

*Quantification of the elongation rate*. For both tubes elongating from buds in vitro or tube elongating in the cellular plasma membrane, the elongation rate was obtained by plotting the length of the tube (increasing with time) at different time points. The slope of the fit to a linear curve directly gives the elongation rate.

**Reporting summary**. Further information on research design is available in the Nature Research Reporting Summary linked to this article.

## Data availability

The authors declare that all quantifications supporting the findings of this study are available within the paper and its Supplementary Information files, and are provided as a Source data file. Other data, such as raw and processed microscopy images, as well as all any other data generated in this study, are available from the corresponding authors on request. Source data are provided with this paper.

## Code availability

The computer codes used for simulations are available as Supplementary Software 1.

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

## Acknowledgements

This work was supported by the Spanish Ministry of Science and Innovation (PGC2018-099645-B-I00 to X.T., PID2019-110949GB-I00 to M.A., BFU2016-79916-P and PID2019-110298GB-I00 to P.R.-C.), the Spanish Ministry of Economy and Competitiveness/FEDER (BES-2016-078220 to C.T.), the European Commission (H2020-FETPROACT-01-2016-731957), the European Research Council (Adv-883739 to X.T., CoG-681434 to M.A.), the Generalitat de Catalunya (2017-SGR-1602 to X.T. and P.R.-C., 2017-SGR-1278 to M.A.), the prize "ICREA Academia" for excellence in research to P.R.-C. and to M.A., Fundació la Marató de TV3 (201936-30-31 and 201903-30-31-32), and "la Caixa" Foundation (Agreement LCF/PR/HR20/52400004). IBEC and CIMNE are recipients of a Severo Ochoa Award of Excellence from the MINECO. We would like to thank all the members of P. Roca-Cusachs, X. Trepat, and M. Arroyo laboratories for technical assistance and discussions. We thank M. Pons, X. Menino, M.G. Parajo, M.-A. Rodriguez, N. Castro, R. Sunyer, I. Granero, V. González, the Unitat de Criomicroscòpia Electrònica (Centres Científics i Tecnològics de la Universitat de Barcelona, CCiTUB), and the MicroFabSpace and Microscopy Characterization Facility, Unit 7 of ICTS "NAN-BIOSIS" from CIBER-BBN at IBEC, for their excellent technical assistance.

## Author contributions

A.L.L.R., C.T., N.W., M.A., and P.R.-C. conceived the study; A.L.L.R, X.Q., M.S., X.T., and P.R.-C. designed the experiments, C.T., N.-W., and M.A. designed the simulation; A.L.L.R., X.Q., D.Z., and M.S. performed the experiments; C.T and N-W. performed the simulation. A.L.L.R., X.Q., and P.R.-C. analyzed the experiments; A.L.L.R., C.T., N-.W., and M.A. analyzed the simulation; and A.L.L.R., M.A., and P.R.-C. wrote the manuscript. All authors commented on the manuscript and contributed to it.

## Competing interests

The authors declare no competing interests.
