## [Peer Review File · Nature Communications]

Reviewer #1 (Remarks to the Author):

Recommendation: Major Revision

In this manuscript, the authors present new observations and dynamics of membrane remodeling due to BAR domain proteins. In a fluorescently labeled supported lipid bilayer system that when stressed can produce bilayer tubes and buds, amphiphysin can constrict existing tubes and elongate bud necks generating tubes and depleting buds. These images are shown in comparison to a recently developed mean field model describing the dynamic feedback between surface density and organization of BAR domains and the bilayer shape. Together, the authors explore the dynamics of reshaping preformed membrane structures, which may be important for how cells respond to external mechanical stress.

Overall, this work presents images of BAR domains reshaping preformed bilayer structures and uses a mean-field model to understand how BAR domains may organize and further reshape the bilayer in response to the existing geometry. It is an important addition to the field's understanding of BAR domains and membrane remodeling in general. However, the current manuscript requires significant revision to clarify the agreement between the experiment and model and to improve its structure and readability.

Comments about text:

1. In the abstract, the authors describe protein arrangement and density as chemical factors. This is confusing because neither of these are related to chemical changes. Indeed, the model developed here does not consider not directly investigate chemical changes to the BAR domain nor the membrane. A more appropriate term should be used such as “molecular”.
2. In the main text, the authors do not use traditional subheadings like Introduction, Results, or Discussion. The authors should use subheadings to improve the readability of their article.
3. In the main text, each paragraph includes reference to several figures, figure panels, and Suppl. figures or videos. To improve readability, the authors should consider breaking up their many paneled figures into multiple figures and introducing panels/figures in sequential order.
4. On page 3, the authors say the tubes expelled from vesicles have diameter of ~25nm (ranging from 10 to 50nm) in Suppl. Fig. 2b. Is there any evidence that the tubes generated on the SLB span a similar range? The authors say that they cannot measure radius in the SLB system. Why is the sucrose case included? Furthermore, is preferred tube diameter recapitulated in the model?
5. In the first paragraph of page 3 and later in the first paragraph of page 6, the authors discuss membrane tearing due to membrane tension. Is it correct to assume that these are the only conditions where tearing should be expected? Further clarification should be added. Additionally, it is unclear how the current model could accommodate tearing events so further discussion ruling out bilayer tear is needed to not invalidate the current model.
6. On page 4, “As per the model, proteins should adsorb on the entire membrane but at a faster rate at the neck, where membrane curvature is more favorable than on the vesicles or on the flat part.” Does the model distinguish between direct binding events or diffusion

from the flat sheet to the neck region? It should be noted again here that protein binding is instantaneous.

7. On page 4, "...close to the radius of tubes scaffolded by Amphiphysin (Suppl. Fig. 2b and 4b)." However, Suppl. Fig. 4b does not show the radius of the tubes scaffolded by Amphiphysin.
8. On page 4, "Consistent with model predictions, the experimental observations systematically captured the growth of thin necks connecting shrinking vesicles to the supported bilayer (Fig. 3a, Suppl. Fig. 4c and Suppl. Video 9)." Can the authors provide an estimate of the vesicle diameter as a function of time to compare to the results of their model?
9. On page 4, "We also found that the higher the concentration, the faster reshaping occurred." It would be very helpful if the author supplied estimates of these timescale and how they differed with the change in concentration or even better surface coverage. Suppl. Fig. 4c attempts to convey this information with unevenly spaced images from a single bud/tube at each concentration.
10. While the authors note on page 5 that the comparison of model and experimental timescales is not possible, Suppl. Fig. 4b suggests the neck elongation even should occur on the timescale of 10^0 seconds while Suppl. Fig. 4c shows neck elongation occurs on the timescale of 10^2 seconds. Are the time axes of Suppl. Fig. 4b incorrect or do the reshaping in the model and experiment differ by 2 orders of magnitude?
11. On page 6, the author write "As a result, initiation of bud elongation or tube pearling occurred at ~ 0.4 and $\sim 0.25-0.35$ coverage respectively, approximately matching theoretical predictions (Suppl. Fig. 7b)". However, based on Suppl. Fig. 7b., the range on this estimate looks to be about 0.3-0.7 surface coverage for bud elongation and 0.2-0.7 for tube pearling. An error estimate would be helpful for the reader to determine if these densities at initiation are statistically significant.
12. On page 6, the authors report the elongation rates in the cellular and in vitro environments and say that the elongation rate is dependent on bulk concentration of amphiphysin. If reasonable, it would be interesting to estimate the "bulk" concentration of amphiphysin required to produce the elongation rates seen in the cellular environment.
13. On page 12, the authors note that all systems are assumed to be axisymmetric. However, none of the experimental images show tubes that are perfectly straight. Further discussion of the validity of this assumption is required considering the noticeable difference between simulation and experiment. Additional discuss related to how this protein diffusion, elongation rates, and other dynamic quantities would also be helpful.
14. Throughout the description of the theoretical model, the authors omit a discussion of thermal fluctuation modes in the bilayer and how that effects shape. Based on their experimental images, the membrane tubes are bent and undulate. How do thermal undulations in bilayer shape affect the organization of proteins and change the dynamics of bilayer reshaping? If it is not important, the authors should discuss why it can be omitted. Membrane fluctuation mediated forces have been shown to be very important for BAR domain assembly on membranes at low coverage, see M. Simunovic, A. Srivastava, and G. A. Voth, "Linear Aggregation of Proteins on the Membrane: A Prelude to Membrane Remodeling", Proc. Nat. Acad. Sci. USA **110**, 20396–20401 (2013). This issue should be discussed and this paper cited.

15. On page 12, the authors discuss the initial bilayer shape prior to protein binding. In the experimental system, there are a variety of shapes for a given condition. Is there a similar heterogeneity present in the simulations?
16. In Eq 6, how is the diffusion of membrane-bound proteins at the boundary accounted for in the protein mass balance?
17. On page 14, great detail is used to describe the model, but there is a lack of emphasis on what developments are new to this particular manuscript.
18. On page 15, the authors detail a fit to the underlying mean field density functional theory. It should be stated very clearly where each model is used.
19. On page 15, the authors suggest various ansatz simplifying their previous mean-field model. Little justification is given for ansatz given in Eq 14 and 16.
20. On page 16 and earlier in the main text, it is noted that bulk diffusion is ignored. Considering the differences in timescales of reshaping between experiment and simulation (Suppl. Fig. 4), some estimate of the bulk transport and how it compares to other noted timescales (e.g., shape dynamics, protein diffusion, protein adsorption) would be helpful. Overall, it is unclear how this assumption affects the validity of their model.
21. On page 17, the authors write “described in Section 4.2 of this supplement.” There is no Section 4.2.
22. On page 17, the authors write “The protrusion can also exchange water with the adhered part of the system...” It is unclear how this is considered in the model and how the energetic penalties and related timescales of such events affect the model results.
23. On page 17, the authors write “because the later is changing” and it should be “latter”
24. It would be valuable to cite this review article in the proper place or places: M. Simunovic, G. A. Voth, A. Callan-Jones, and P. Bassereau, “When Physics Takes Over: BAR Proteins and Membrane Curvature”, *Trends Cell Biol.* **25**, 780-792 (2015).

Comments about figures:

25. In Fig. 2b-e, the energy density landscapes are “according to our mean field density functional theory”. Is it correct to assume these energy density landscapes were calculated using Eq. 11 and not the explicit parameterization given by Eq. 12,15? This should be clarified in caption and in text.
26. In the caption of Fig. 2b-e, the plots are described as energy density landscapes. However, these are free energies and should be described in agreement with Suppl. Fig. 3 and the in-text description on page 15.
27. In Fig. 2b-e, are the depictions of the isotropic and nematic phases to scale (i.e., is the protein length proportional to the membrane size?)?
28. In Fig. 2b-e, it is unclear how the model accounts for the “crowding effects” that render surface coverage greater than 0.7 impossible. It is unclear from figure caption and theory section.
29. In Fig. 2d,e, the proteins in state i are shown dispersed and aggregated, respectively. The proteins in states ii and iii are shown distinctly different ordering with respect to each other (offset or in a line) and with respect to the long axis of the tube (perpendicular or tilted). It is unclear what aspects of these depictions (and those in Suppl. Fig. 3) are based on model data or hypotheses and could be more misleading than helpful.
30. In Fig. 2f, there is text that is cutoff. “Chemical distributio”

31. Fig. 3a compares simulation and experiments showing bud reshaping. However, the simulation does not have a timescale associated with it.
32. Fig. 3f is a model (left) and experimental (right) timeseries of tube reshaping.
 - a. However, the end points of the tube reshaping are qualitatively different. The model has pearls remaining while the experimental system does not appear to. It is unclear why this occurs. Does the model overstabilize the pearl? Is the effective protein coverage lower compared to the experiment?
 - b. Fig. 3f left does not have any estimate of time. What is the model time of the series of snapshots?
33. In Fig. 3j, the concentrations should be in μM for consistency.
34. In Fig. 3j right, experimental images of pressurized caps are shown. An explanation of how these images differ from the bud structures previously shown would be helpful.
35. The caption of Suppl. Fig. 2c states “but no reshaping in the form of thin tubes is observed.” However, there is no estimation of the tube radius. How do the authors determine that the tubes are not thin nor become thin and what is the definition of thin used here?
36. In the caption of Suppl. Fig. 3, describes the landscapes as “landscape of the free-energy density” instead of “energy density landscape”. Is it correct to assume these are interchangeable terms?
37. Suppl. Fig. 4a shows a timeseries of vesicle consumption. When comparing the final structure of Suppl. Fig. 4a. to structures in the phase diagram of Suppl. Fig. 4b, it appears that the final structure is not the equilibrated structure because the bud in the final panel is a tear-drop shape rather than nearly-spherical. It would helpful to add to this timeseries as the final relaxation occurs.
38. Suppl. Fig. 5a. “Mean Intensity Protein = $RI_{\text{lipid}} / A_{\text{protein}}$ ” What is RI_{lipid} and how does it differ from RI_{bilayer} ?
39. Suppl. Fig. 6d shows a timeseries without units of time and shows an end state of many disconnected pearls not seen in Suppl. Fig. 6c. It is unclear that this is a stable state or an intermediate toward an elongated tube. The caption also omits the bulk concentration of amphiphysin and the reader cannot compare the unlabeled control to the labeled experiments. Overall, the similarity between unlabeled control and labelled experiments compare.
40. In Suppl. Fig. 7b, what is the difference between the variations of orange and green colors? Is there a significance to the size of the markers in the left and right of Suppl. Fig. 7b?

Reviewer #2 (Remarks to the Author):

The manuscript by Le Roux et al. and Roca-Cusachs examines membrane shape changes induced by elongated curved proteins, commonly known as BAR domain proteins. The authors use lipid bilayer membranes deposited onto activated (oxidized) PDMS surfaces that can be stretched so as to increase area available for liposomes to fuse. This membrane excess area can be consumed by the formation of membrane tubules upon stress relaxation in the PDMS sheet. The authors study interactions of the BAR protein amphiphysin with this system.

In a second thrust, the authors developed a theory that allows to study disorder-nematic ordering transitions of curved proteins on membrane substrates. The authors find interesting couplings between the shape of the membrane and phase diagrams that separate nematic from disordered states as a function of density of protein on the membrane, and membrane shape.

In a third thrust, the authors put cells onto their substrates and find similar curvature transitions at the plasma membrane as those in the model membrane system.

While the overall scope of the study is impressive, I am not convinced that the experimental results significantly add to the existing literature, which is barely discussed in the context of the authors' findings. The experimental results appear to be somewhat disconnected from the theory insight. For example, experiments do not analyze nematic ordering of proteins on the membrane, a key prediction of the calculations. The project therefore seems to be preliminary in as far as the junction between theory and experiments is concerned. I have the following more specific comments for the authors to consider:

Major:

1) The authors find in their theory, that for a wide range of bud diameters and protein concentrations, the enrichment on tubular regions versus bud regions was approximately two-fold. This result is confirmed by experiments. I believe that this result may contradict early studies on the curvature sorting of proteins on membranes that are available in the literature. The result also appears to contradict a recent study by Larsen et al. and Stamou, ACS Central Science 2020, which is not cited.

2) The bilayers formed on the PDMS substrates look highly heterogeneous, suggesting that a complex curvature pattern may exist in those-substrate supported bilayers. It is not clear how that complication would affect the comparison between theory and experiments. Furthermore, solid supported lipid bilayers may experience asymmetry regarding their trans-bilayer stress distribution, due to interactions with the substrate. For example, it has been shown that solid supported membranes develop asymmetric lipid compositions. How would such effects factor into experimental observations as well as comparison between theory and experiments?

3) The authors mention that they did not control the dynamics of protein delivery to their membranes. However, it could be argued that microfluidics has become sufficiently ubiquitous to

claim that the experimental design used here may not be state-of-the art. It seems to me that any experiment describing dynamic aspects of protein-lipid interactions ought to carefully control the dynamics of protein delivery in order to be able to produce meaningful and reproducible data.

4) The cellular experiments aim to show that the imposition of membrane curvature can lead to the triggering of BAR protein responses. Similar coupling between imposed curvature and intra-cellular proteins has previously been described, e.g. by Galic et al. and Meyer, Nat Cell Bio 2012. It is not clear to what extent the cellular experiments shown here move significantly beyond the existing literature.

Minor:

1) The authors use a somewhat unusual lipid composition that considers the headgroups PC, PS, and PA. I am not aware that PA is present in the plasma membrane to significant amounts. Why not work with a physiologically more relevant lipid mixture?

2) The control of adding neutravidin, which is not observed to bind to the bilayer, would have been more useful if done with a protein that does bind, but that is not supposed to generate membrane curvature. What is the addition of a non-binding molecule supposed to test? It does not seem that this control was motivated, or the results put into context.

3) It is unclear to what extent the authors considered protein crowding effects and their coupling with membrane curvature in their model.

4) It would be helpful to clearly lay out what parameters of the theory are experimentally known, and which ones are unknown.

Reviewer #3 (Remarks to the Author):

Review of "Dynamic Mechanochemical Feedback Between Curved Membranes and BAR Protein Self-Organization"

Le Roux et al. demonstrate that the BAR domain-containing protein Amphiphysin binds to and induces morphological changes in lipid bilayer nanotubes. This study introduces a new experimental technique with control over membrane tension for producing and imaging supported lipid bilayers.

Using this system, the authors first stretch and then relax membranes to generate low-tension and low-curvature membrane “templates.” The authors then monitored Alexa488-Amphiphysin binding to these templates over time. Amphiphysin bound to and stabilized bilayer nanotubes and, at high concentrations, remodeled them into pearled tubes and thin necks. The authors complemented these observations with dynamical models that account for the energetics of membrane bending, protein binding, and isotropic versus nematic protein ordering. Their model predicted that, over time, Amphiphysin would concentrate on high-curvature membrane cylinders versus lower curvature vesicles, leading to pearling. Mechanistically, the prediction is that Amphiphysin-induced membrane remodeling emerges from an isotropic-to-nematic transition of the bound protein molecules. The free energy driving this re-ordering arises from the spontaneous and anisotropic curvatures of Amphiphysin. Experimentally, this prediction may have been verified by measuring the relative intensity of the protein coat on high curvature tubes versus low curvature vesicles. Finally, using the same stretch-and-release technique, the authors demonstrate that analogous membrane tubes form from the plasma membrane of living cells. Like the *in vitro* tubes, plasma membrane tubules appear to recruit over-expressed GFP-Amphiphysin, providing a mechanism for mechanochemical signal transduction in cells.

Overall, this work utilizes a new experimental setup to generate low-tension lipid bilayer tubules and uses this system to explore membrane remodeling dynamics by a well-studied protein. The theoretical model, which is being reviewed in a separate manuscript and provides mostly intuitive results, may be overly simplistic to offer novel insights. Additionally, the concept of an isotropic-to-nematic transition for Amphiphysin is not novel and can be found in the first reference and other sources. Overall, we believe that this manuscript requires major and minor revisions before being reconsidered for publication in *Nature Communications*.

Major concerns:

1. The experiments relied on confocal fluorescence microscopy. Unless we are mistaken, the data presented were all single Z slices, rather than Z-stacks or volumes. We are concerned that without sampling in the Z dimension, the quantifications may be systematically in error due to the imaged structures' unknown morphology. For example, a nanotube protruding in the Z direction will appear as a dome or vesicle. Similarly, the morphology of the structures at the end of the bilayer necks cannot be unambiguously identified as spheres from single Z slices. Z stacks should be included in this manuscript to establish the morphologies assigned to the bilayer structures firmly.

2. Further, please include more information about the microscopy in the methods section. For example, how are the Z heights for the micrographs in the Figures optimized? This concern is especially critical for quantifying the fluorescence from tubes versus buds and vesicles.

3. Most of the interpretation of the experimental data relies on a theoretical model. A manuscript describing this model is under review separately. For this review, we assume that the model proves sound. Given the importance of the modeling results to this manuscript, we think this manuscript should have more information about the model and its limitations or that the acceptance of this manuscript should be contingent on the approval of the complete description of the model. Pertinent questions include: Does the model contain attractive interactions between protein molecules? If so, are these interactions orientation-dependent? Prior work indicates that BAR domain proteins self-assemble into curvature- and orientation-dependent lattices. Are interactions between the charged membrane surface and the protein explicitly or implicitly included in the model? What are the limitations of using continuum models for highly curved bilayers? When do molecular effects such as induction of asymmetry between the leaflets become significant? Overall, our concern is that the model may be overly simplistic for a phenomenon as complicated as membrane remodeling by full-length Amphiphysin.

4. The authors connect their new experimental setup and model to biology by examining GFP-Amphiphysin binding to plasma membrane tubes in cells and report this as a novel form of mechanosensing and signal transduction—but this idea is not as novel as presented in the manuscript. Contributions to the concept of membrane tension-regulated endocytosis include:

Dai, J., Sheetz, M.P., Wan, X., and Morris, C.E. (1998). Membrane tension in swelling and shrinking molluscan neurons. *J Neurosci* 18, 6681-6692.

Raucher, D., and Sheetz, M.P. (1999). Membrane expansion increases endocytosis rate during mitosis. *The Journal of Cell Biology* 144, 497-506.

Dai, J., Ting-Beall, H.P., and Sheetz, M.P. (1997). The secretion-coupled endocytosis correlates with membrane tension changes in RBL 2H3 cells. *J Gen Physiol* 110, 1-10.

Riggi M, Bourgoint C, Macchione M, Matile S, Loewith R, Roux A. (2019) TORC2 controls endocytosis through plasma membrane tension. *J Cell Biol.* 218, 2265-2276.

5. Some BAR domain proteins can associate with membranes via more than one surface (Frost et al. 2008), and work has shown that Amphiphysin, in particular, acts through more complex mechanisms than simple induction of spontaneous curvature through its crescent-shaped BAR domain. Physiochemical effects such as amphipathic motif insertion, phase separation, and steric crowding by low complexity domains play increasingly appreciated roles in membrane remodeling in reconstituted systems and cells. See, for example:

Day KJ, Kago G, Wang L, et al. (2019) Liquid-like protein interactions catalyze the assembly of endocytic vesicles. *bioRxiv*.

Snead WT, Zeno WF, Kago G, Perkins RW, Richter JB, Zhao C, Lafer EM, Stachowiak JC. (2018) BAR scaffolds drive membrane fission by crowding disordered domains. *J Cell Biol.* 218, 664-682.

These ideas merit discussion in this manuscript, especially in whether and how they contribute to the theoretical modeling. Please discuss how such effects inform the model of Amphiphysin-induced remodeling within a broader overall discussion of their results concerning the copious literature regarding the action of BAR domain proteins.

Minor concerns:

1. In the micrographs of amphiphysin-GFP coated tubes, why is the membrane marker visible on membrane caps/vesicles but not the membrane tubes?
2. Many references are missing information or improperly formatted.

Response to reviewer comments for manuscript “Dynamic Mechanochemical feedback between curved membranes and BAR protein self-organization”.

Reviewer comments marked in black, responses marked in blue.

Reviewer #1 (Remarks to the Author):

Recommendation: Major Revision

In this manuscript, the authors present new observations and dynamics of membrane remodeling due to BAR domain proteins. In a fluorescently labeled supported lipid bilayer system that when stressed can produce bilayer tubes and buds, amphiphysin can constrict existing tubes and elongate bud necks generating tubes and depleting buds. These images are shown in comparison to a recently developed mean field model describing the dynamic feedback between Surface density and organization of BAR domains and the bilayer shape. Together, the authors explore the dynamics of reshaping preformed membrane structures, which may be important for how cells respond to external mechanical stress.

Overall, this work presents images of BAR domains reshaping preformed bilayer structures and uses a mean-field model to understand how BAR domains may organize and further reshape the bilayer in response to the existing geometry. **It is an important addition to the field’s understanding of BAR domains and membrane remodeling in general.** However, the current manuscript **requires significant revision to clarify the agreement between the experiment and model and to improve its structure and readability.**

We thank the reviewer for his/her relevant comments and questions. We have carried out simulations in broader conditions to better address the agreement between model and experiment. In particular, we have now a much better understanding of the timescales involved. We also want to emphasize that once this is established, we can use the model to complement our understanding of the experimental data, since some aspects are very challenging to measure experimentally, as for instance the ultrastructure of the bilayer template. We also structured the paper in paragraphs to improve its readability. We have separated it in the subparts as described in responses below.

Comments about text:

1. In the abstract, the authors describe protein arrangement and density as chemical factors. This is confusing because neither of these are related to chemical changes. Indeed, the model developed here does not consider not directly investigate chemical changes to the BAR domain nor the membrane. A more appropriate term should be used such as molecular.

We thank the referee for this comment and have rephrased the abstract substituting “chemical” by “molecular organization”.

2. In the main text, the authors do not use traditional subheadings like Introduction, Results, or Discussion. The authors should use subheadings to improve the readability of their article.

We have separated the text with the following subheadings, which we hope guides the reader in a clear manner:

- Introduction
- A novel system to study the dynamics of curved membrane remodeling by BAR proteins
- Reshaping is governed by a low-curvature isotropic to high-curvature nematic phase transition with coexistence of both states.
- Dynamic reshaping of buds and tubes
- Physical parameters governing the reshaping of mechanically-induced curved lipid templates
- Reshaping is hindered by high tension
- Cellular compression triggers the formation of Amphiphysin-rich lipid tubes
- Conclusion

3. In the main text, each paragraph includes reference to several figures, figure panels, and Suppl. figures or videos. To improve readability, the authors should consider breaking up their many paneled figures into multiple figures and introducing panels/figures in sequential order.

We have simplified Fig 3 which was crowded. Supplementary figures also were redesigned according to the new results. We have revised the way we introduce the figures and cite them sequentially as the text goes.

4. On page 3, the authors say the tubes expelled from vesicles have diameter of ~25nm (ranging from 10 to 50nm) in Suppl. Fig. 2b. Is there any evidence that the tubes generated on the SLB span a similar range? The authors say that they cannot measure radius in the SLB system. Why is the sucrose case included? Furthermore, is preferred tube diameter recapitulated in the model?

As the reviewer points out, the SLB system does not allow for a sufficiently accurate measurement of tube radius. We included the sucrose case as a control, to confirm protein activity and verify accordance with results obtained in the field with Amphiphysin (for instance in Takei, et al, and De Camilli, Nat. Cell Biol., 1999, 10.1038/9004 or Peter, et al, and Mc Mahon, Science, 2004, 10.1126/science.1092586). In this assay, proteins are incubated at high concentration with vesicles, and samples are treated when the equilibrium state is reached. Electron microscopy has extensively characterized how BAR proteins oligomerize around the tube and scaffold it. We acknowledge that we do not know if the long thin tubes reshaped in our curved templates will have the same exact diameter as in the vesicle assay. In fact, the preferred curvature imposed by Amphiphysin most probably depends on the initial mechanical state of the lipid template used and on other chemical factors as the protein to lipid ratio, the lipid composition or the ionic strength of the buffer. Different results coming from such experiments or in tube pulling assays can be found in the literature. In our set up, we do not pretend to determine the

exact diameter of the reshaped tube as this is not possible, but to describe the physical principles governing the dynamics of the reshaping process. We now rephrased the paragraph mentioning the sucrose loading assay earlier in the first paragraph of the results [section A novel system to study the dynamics of curved membranes remodeling by BAR proteins] to clarify its purpose.

Regarding the theoretical model, the preferred tube diameter is largely determined by a model parameter encoding the intrinsic curvature of molecules. As such, it does recapitulate the tendency of tubes to adopt a preferred diameter if coverage is large-enough, but this parameter is precisely chosen to match the known preferred diameter from the literature, and thus is not a model prediction. We have rephrased the text in the results [section Dynamic reshaping of tubes and buds] to convey the idea that the model prediction is that at high-enough coverage the tube diameter tends to the protein preferred curvature.

5. In the first paragraph of page 3 and later in the first paragraph of page 6, the authors discuss membrane tearing due to membrane tension. Is it correct to assume that these are the only conditions where tearing should be expected? Further clarification should be added. Additionally, it is unclear how the current model could accommodate tearing events so further discussion ruling out bilayer tear is needed to not invalidate the current model.

The model used as a conceptual background in the experiments does not explicitly describe tearing but allows us to estimate membrane tension. As previously examined in Staykova, et al, and Stone, Phys Review Letters, 2013, 10.1103/PhysRevLett.110.028101, in the absence of proteins, the bud-like and tubular protrusions off a SLB have very low tension. In contrast, the shallow spherical caps obtained for little excess area but large excess volume are highly tense and it was shown previously that upon osmotic or stretch perturbations these shallow caps could lyse. Here, the model predicts that the addition of proteins does not induce significant reshaping of shallow caps, as expected for a tense structure with little excess area, but increases membrane tension up to values higher than lysis tension, and this is confirmed by our experiments, which show tearing of shallow membrane protrusions upon protein exposure. We have clarified this point in the revised manuscript [section Reshaping is hindered by tension].

6. On page 4, “As per the model, proteins should adsorb on the entire membrane but at a faster rate at the neck, where membrane curvature is more favorable than on the vesicles or on the flat part.” Does the model distinguish between direct binding events or diffusion from the flat sheet to the neck region? It should be noted again here that protein binding is instantaneous.

The model accounts for both adsorption from the bulk and diffusion on the membrane. In the model, protein binding is not instantaneous but rather follows the adsorption kinetics described in [Theoretical model Section 2 Modeling the protein-membrane interaction dynamics]. As the referee suggests, neck enrichment may have two contributions: direct adsorption from the bulk, which is faster at the neck than in neighboring regions due to a more favorable curvature, and diffusive transport from adjacent membrane regions. Since adsorption rate is size-independent but diffusive transport is not, the former should be faster beyond a length-scale. However, it is unclear *a priori* if during the dynamics,

differences in chemical potential of proteins on the membrane provide a significant driving force for diffusion towards the neck. Indeed, while the mechanical part of the chemical potential is favorable at the highly curved neck, the entropic part is not when this region is enriched. Thus, the question asked by the referee is not trivial. To address it, we quantified in our simulations the fraction of protein delivery to the growing neck due to adsorption, and found that it is overwhelmingly dominant. This is now discussed in the main text [section Dynamic reshaping of tubes and buds] and in the Theory Supplement (section 7).

7. On page 4, "...close to the radius of tubes scaffolded by Amphiphysin (Suppl. Fig. 2b and 4b)." However, Suppl. Fig. 4b does not show the radius of the tubes scaffolded by Amphiphysin.

We agree with the reviewer that the sentence was confusing, and as we rephrased the paragraph related to the tube diameter in the context of the sucrose loading assay/electron microscopy, this sentence has in fact been removed.

8. On page 4, "Consistent with model predictions, the experimental observations systematically captured the growth of thin necks connecting shrinking vesicles to the supported bilayer (Fig. 3a, Suppl. Fig. 4c and Suppl. Video 9)." Can the authors provide an estimate of the vesicle diameter as a function of time to compare to the results of their model?

We thank the reviewer for this interesting suggestion. Indeed, watching the video of bud reshaping clearly shows this "lipid consumption" and a decrease in bud diameter as the tube elongates from its neck. Though the vesicle moves as the tube elongates to quite extensive lengths, we can identify some time frames at which the vesicle clearly appears in a round shape and measure its diameter as time is passing; the corresponding quantification has been added to Suppl. Fig 4b.

In terms of the model, we fixed an initial diameter and adhesion potential and plotted the diameter of the elongating buds over time. In Figure S4b, we show both experimental quantification and outcome from the simulations, and confirmed in both cases the decrease in diameter with time, indicating that as the tube elongates, it consumes lipids from the vesicle. Since the volume of fluid enclosed in a bud is much larger than that in a thin tube of equivalent area, the process of neck elongation and bud shrinkage necessarily requires exchange of enclosed volume and/or membrane area between the protrusion and the surroundings. More generally, the area/tension and volume/pressure mechanical ensemble to which an evolving protrusion is subjected to should affect the reshaping dynamics. In the revised manuscript, we examine more extensively the robustness of our results with respect to the mechanical ensemble, Section 8 in the Theory Supplement and Supp. Figs. 7, 8 and 9c. Regarding bud consumption, we find that it requires some degree of volume exchange between the protrusion and its surrounding. We now included these analyses in the text [section Dynamic reshaping of tubes and buds] and the plot in Supp Fig 4b.

9. On page 4, "We also found that the higher the concentration, the faster reshaping occurred." It would be very helpful if the author supplied estimates of these timescale and how they differed with the change in concentration or even better surface coverage. Suppl.

Fig. 4c attempts to convey this information with unevenly spaced images from a single bud/tube at each concentration.

We thank the referee for this and other comments about timescales. In the revised manuscript, we have better pinpointed the mechanism governing the different rates of reshaping dependent on concentration, see Theory Supplement (Section 7) and Suppl. Figs. S5 and S6. In the simulations of the original manuscript, we instantaneously modified the bulk protein concentration and followed the dynamics resulting from adsorption, diffusion of proteins on the membrane and reshaping dynamics dragged by membrane viscosity. Now, accounting for the injection process in our experiments we have found that bulk diffusion in the medium droplet between the injection point and the SLB is in fact the slowest phenomenon in our experiments. As a result of this scale-separation, a protrusion will equilibrate very fast in response to a change in the concentration of proteins to which it is locally exposed, which occurs at a slower rate through diffusion in the bulk. Analyzing the dynamics under different nominal concentrations provides in fact a neat control of this physical picture as a given local protein concentration at the SLB is reached at different timepoints. The model and these controls show that, as the referee suggests, coverage is a key parameter that controls reshaping. Coverage rapidly equilibrates by adsorption given the local protein concentration in solution, whose dynamics is determined by slow bulk diffusion. Thus, we can predict the time required for the critical coverage to trigger reshaping depending on the nominal concentration.

Accordingly, we inserted this diffusion analysis in Suppl. Fig. S5 and S6 and re-purposed the snapshots of the former figures, to illustrate the mismatch in timescales between theory and experiment attributed to diffusion. As a consequence, we also removed the dynamical diagrams from Fig 3. We now address this issue in the text [section Physical parameters governing the reshaping of mechanically-induced curved lipid templates] starting with the observation that the protein binding curves of Fig 3 show that bud elongation or tube pearling occur at a protein fluorescent intensity measured on the curved template (proportional to coverage). Further, In Suppl. Fig S9 we quantify protein coverage when reshaping starts for several tubes and buds (by either pearling or elongating, respectively). As the reviewer asks in another question, we now also added statistics to the coverage quantifications and mentioned in the manuscript the means and standard deviation of the estimates obtained.

10. While the authors note on page 5 that the comparison of model and experimental timescales is not possible, Suppl. Fig. 4b suggests the neck elongation even should occur on the timescale of 10^0 seconds while Suppl. Fig. 4c shows neck elongation occurs on the timescale of 10^2 seconds. Are the time axes of Suppl. Fig. 4b incorrect or do the reshaping in the model and experiment differ by 2 orders of magnitude?

As discussed in the previous point, following the comments of this and of other referees, we have revisited more carefully the comparison of the dynamics between theory and experiments. As discussed in the new Theory Supplement, Section 7, the slowest dynamical process is the protein diffusion from the injection point to the SLB, which can take up to a few minutes to reach a significant fraction of the nominal protein concentration. In comparison, a protrusion exposed to a change in medium protein concentration equilibrates mechanically and chemically within a few seconds. In the revised manuscript we estimate the time-evolution of protein concentration to which the

SLB is effectively exposed by solving a diffusion equation in the medium. The solution of the diffusion equation provides a mapping between time and concentration at the SLB depending on the nominal concentration. As described in the previous answer, this new understanding has affected the way we present our results in Figure 3, where we focus on the mechanistic steps of reshaping and on the critical coverage for reshaping. Suppl. Figures 5 and 6 discuss how bulk diffusion governs the reshaping timescale in our experiments, which is further discussed in Theory Supplement, Section 7.

11. On page 6, the author write “As a result, initiation of bud elongation or tube pearling occurred at ~ 0.4 and $\sim 0.25-0.35$ coverage respectively, approximately matching theoretical predictions (Suppl. Fig. 7b).”. However, based on Suppl. Fig. 7b., the range on this estimate looks to be about 0.3-0.7 surface coverage for bud elongation and 0.2-0.7 for tube pearling. An error estimate would be helpful for the reader to determine if these densities at initiation are statistically significant.

To address this issue, we have now calculated more accurate experimental estimates of coverage at the onset of either pearling or elongation. To this end, we have first estimated the experimental coverage at the onset of reshaping for each analyzed structure. For this, we have fitted the data with an exponential curve, using the equation: $Coverage = C \cdot (1 - e^{-k \cdot t})$ to the experimental evolution with time of each analyzed structure, and taken the coverage value at which reshaping begins. Then, we have calculated the mean and standard deviation of all points. In the case of bud elongation, we obtain a mean coverage of 0.442 with a confidence interval 0.339-0.545 (Standard deviation 0.097, Standard error of the mean 0.04). In the case of tube pearling, we obtain a mean coverage of 0.342 with a confidence interval 0.257-0.427 (Standard deviation 0.08, Standard error of the mean 0.033). In parallel, we present a new sensitivity plot derived from our theory, showing the coverage predicted at the onset of pearling or elongation. This is obtained in different conditions as a function of membrane-support interaction, and different boundary conditions at the endpoint of the membrane, for protrusions of different diameters.

For bud elongation, our experimental quantification falls in the range of coverage obtained from the different outcomes of the simulations. Regarding tubes, theoretical values are somewhat below experimental measurements. This is likely due to the experimental difficulty of precisely capturing the onset of pearling (in contrast to bud elongation, which is a much more obvious event). Indeed, comparing the coverage at the exact onset of pearling with the simulations results is challenging, since visible reshaping in experiments is only clearly observed after it begins from a theoretical point of view. We have now corrected the text [section Physical parameters governing the reshaping of mechanically-induced curved lipid templates] and the methods with the analysis mentioned above.

12. On page 6, the authors report the elongation rates in the cellular and in vitro environments and say that the elongation rate is dependent on bulk concentration of amphiphysin. If reasonable, it would be interesting to estimate the “bulk” concentration of amphiphysin required to produce the elongation rates seen in the cellular environment.

As the reviewer points out, it is extremely challenging to provide such an estimate. To observe the location of Amphiphysin in the cell, we overexpress fluorescent Amphiphysin using a transient transfection and obtain a population of fluorescent cells with a range of bulk concentrations. We can clearly see from the fluorescent image that the protein

distribution is uneven, with a significant amount already bound to the membrane. We note that even if we would provide an estimate, it would probably not reflect the local concentration nor would it be a physiological estimate as we work in overexpression conditions. It is tempting to compare the values of the elongation rates obtained in-vitro at different concentrations (at 0.25 and 0.35 μM injection concentrations, we obtain rates between 20 and 75 nm/s, while at 0.5 μM , we reach rates from 365 to 550 nm/s). The quantification of the cellular elongation rates gave values between 135 and 365 nm/s. However, we cannot directly compare diffusion and adsorption of Amphiphysin to the membrane in the in-vitro system and in cells and this prevents us from obtaining an estimate of Amphiphysin concentrations in cells, even locally.

13. On page 12, the authors note that all systems are assumed to be axisymmetric. However, none of the experimental images show tubes that are perfectly straight. Further discussion of the validity of this assumption is required considering the noticeable difference between simulation and experiment. Additional discussion related to how this **affects(?)** protein diffusion, elongation rates, and other dynamic quantities would also be helpful.

Indeed, our model assumes that the system is axisymmetric but experimental images show fluctuations of tubes. To assess the validity of our assumption, in Methods, Theoretical Model, Section 9, we estimate the persistence length of thin tubes, which we find to be of about 1 μm for the thinner tubes in agreement with our observations. From these calculations, we can estimate the relative local areal change in the inner/outer parts of the bent tube to be below 2% and hence they should lead to small protein coverage fluctuations. We thus can assume that these shape changes will have a mild effect on the dynamics of protein adsorption and transport and on the mechanics of the membrane.

14. Throughout the description of the theoretical model, the authors omit a discussion of thermal fluctuation modes in the bilayer and how that effects shape. Based on their experimental images, the membrane tubes are bent and undulate. How do thermal undulations in bilayer shape affect the organization of proteins and change the dynamics of bilayer reshaping? If it is not important, the authors should discuss why it can be omitted. Membrane fluctuation mediated forces have been shown to be very important for BAR domain assembly on membranes at low coverage, see M. Simunovic, A. Srivastava, and G. A. Voth, “Linear Aggregation of Proteins on the Membrane: A Prelude to Membrane Remodeling”, *Proc. Nat. Acad. Sci. USA* **110**, 20396–20401 (2013). This issue should be discussed and this paper cited.

We thank the referee for raising this issue. We acknowledge that our axisymmetric model does not capture shape fluctuations of the membrane tubes and argue that given the relation between persistence length and radius of membrane tubes, these fluctuations should induce small fluctuations in mechanical strains and protein coverage. The interesting reference alluded by the referee is one in which protein self-assembly breaks the symmetry of an initial planar, isotropic and homogeneous state. In many of the events of reshaping and molecular re-organization reported in our manuscript, the initial membrane template strongly directs the process, e.g. the thin neck of buds, and therefore, we expect that the effect of thermal fluctuations will be small. However, we acknowledge that the pearling instability involved in the reshaping of tubes breaks symmetry, and hence, fluctuations should affect such pattern formation. As we point out in the revised manuscript, other uncertainty factors such as the mechanical ensemble also affect our

estimation of the critical protein coverage for reshaping. Altogether, as acknowledged in the revised manuscript, we expect that thermal fluctuations will not change the physical picture put forth in the present study, but they may re-scale some of the parameters in the model or modify the critical coverage for the pearling instability. We have included a new section (9) in Methods, Theoretical Model discussing this, and discuss it in the main text [section Dynamic reshaping of tubes and buds].

15. On page 12, the authors discuss the initial bilayer shape prior to protein binding. In the experimental system, there are a variety of shapes for a given condition. Is there a similar heterogeneity present in the simulations?

The formation of protrusions in supported lipid bilayers can be explained by a simple conceptual framework according to which the shape of protrusions minimizes the bending energy for a given excess area or excess enclosed volume. This theory provides a framework to control the shape of these protrusions depending on the applied compression or osmotic difference. For instance, to transform buds into longer and thinner tubes, compression and osmotic strength in the medium must be increased. This framework has been previously tested in synthetic systems [Staykova, et al, and Stone, Phys Review Letters, 2013, 10.1103/PhysRevLett.110.028101] and in cells [Komalska, et al, and Roca-Cusachs, Nat. Comm., 2015, 10.1038/ncomms8292]. Thus, through the magnitude of stretch and the medium osmolarity, we have some degree of control on the shape of protrusions. Having said this, in a given sample there is some degree of heterogeneity in the shape of protrusions, likely due to variations in the interaction with the substrate (pinning points, non-uniform surface treatment, etc), which might affect the lateral membrane flow, or the flow of water underneath the membrane. In any case, this variability is interesting since it provides a variety of templates for proteins to act upon. When we wrote “All of these protrusions are observed in our experimental system” we meant in different experiments with different conditions.

As discussed in [Theoretical model Section 1 “Modeling the state prior to protein exposure”], in the model we controlled the excess membrane area and excess enclosed volume to control the shape of protrusions. To address the variability of shapes, we choose a set of representative protrusions and studied the effect of proteins on them in isolation, as a simplification of the real system. In the revised manuscript, we considered three different bud diameters and three different tube lengths to assess the generality of our conclusions (Suppl. Video 17 and 18). Further, in the new Suppl. Fig. 9c we quantify the dependence on protrusion size of the critical protein coverage for reshaping. This figure also looks at the dependence on the mechanical ensemble of a protrusion, as discussed in the Supplementary Theory note, Section 8. This analysis points out the robustness of the model and its validity for the diversity of initial curved templates obtained experimentally. (Suppl. Fig 7).

16. In Eq 6, how is the diffusion of membrane-bound proteins at the boundary accounted for in the protein mass balance?

As mentioned above, we study each protrusion in isolation. Proteins are delivered to the system by adsorption from the bulk governed by Eqs. 6 and 7 (see also the Supplementary Theory note). Then, bound proteins can diffuse on the membrane. In Eq. 6, w is the diffusive flux given by Fick’s law on the membrane. As discussed in response #6, most of the protein delivery to the protein-rich domains is by adsorption from the bulk. In the

revised manuscript, we have additionally accounted for diffusion in the bulk, see Methods, Theoretical Model, Section 7.3.

17. On page 14, great detail is used to describe the model, but there is a lack of emphasis on what developments are new to this particular manuscript.

As described and cited in the manuscript, the modelling relies on accompanying manuscript, which at the time of the original submission was in Arxiv. This manuscript is now published (Tozzi C., et al., et Arroyo, *Soft Matter*, 2021, 10.1039/d0sm01733g), so we can now streamline the presentation of this model in the main text and in the Method part, Theoretical Model. This previous work provided a mean-field theory for curved proteins on curved surfaces of *fixed* shape. In this manuscript, we combine this theory with a dynamical model for membrane reshaping, allowing us to describe the two-way interplay between membrane shape and molecular organization.

18. On page 15, the authors detail a fit to the underlying mean field density functional theory. It should be stated very clearly where each model is used.

We have further emphasized this point in Section 4.2 of Methods, Theoretical Model.

19. On page 15, the authors suggest various ansatz simplifying their previous mean-field model. Little justification is given for ansatz given in Eq 14 and 16.

As we now describe, Eqs. 14 and 15 can be easily justified using algebraic manipulations and one approximation, the so-called Doi closure. As for Eq. 16, we found that 15 was not enough to fit the mean field theory and we proposed a purely phenomenological additional term in the ansatz. We do not attach to the form of this functional a precise physical meaning other than its ability to accurately parametrize the mean field theory. Physically motivated ansatz would be preferable from a theoretical point of view but this is beyond the scope of the present work. We have clarified this aspect of the model, in Methods, Theoretical Model.

20. On page 16 and earlier in the main text, it is noted that bulk diffusion is ignored. Considering the differences in timescales of reshaping between experiment and simulation (Suppl. Fig. 4), some estimate of the bulk transport and how it compares to other noted timescales (e.g., shape dynamics, protein diffusion, protein adsorption) would be helpful. Overall, it is unclear how this assumption affects the validity of their model.

We thank the referee for this comment, which helped us better understand the dynamics experimentally observed, as discussed in previous points. Having established that the dynamics of bulk diffusion are the slowest in our system, and hence control the dynamics in our experiments, we can use this fact to test our model, according to which protein coverage on the membrane is the key physical parameter that controls reshaping. Indeed, for a given nominal protein concentration, we can predict the time at which a given chemical potential of dissolved proteins is attained at the SLB, and since equilibration of dissolved and adsorbed proteins is fast, we can predict the time it takes to reach a critical coverage for reshaping. Since this time depends on the nominal concentration, we can compare model and experiment to test our predictions. Taking bulk diffusion into account addresses the time scale difference as studied in Suppl. Figs 5 and 6, which confirms the robustness of our model.

21. On page 17, the authors write “described in Section 4.2 of this supplement.” There is no Section 4.2.

The reviewer is correct and the Section was 3.2. But now we revised our Theoretical Supplement and the section actually corresponds to 4.2 "Explicit parametrization of the theory".

22. On page 17, the authors write “The protrusion can also exchange water with the adhered part of the system...” It is unclear how this is considered in the model and how the energetic penalties and related timescales of such events affect the model results.

We thank the referee for this comment. Since we study each protrusion in isolation, it is essential to fix the ensemble that controls how our protrusions exchange membrane or enclosed fluid with the rest of the system. Unfortunately, this aspect cannot be characterized experimentally at this point. Furthermore, in different systems where our conceptual framework may be applicable, the ensemble may be very different depending on area availability in nearby reservoirs or on the permeability of the interstitial space. For this reason, we chose to vary the ensemble and test the generality of our results. For membrane exchange, we interpolated between fixed tension (allowing for membrane exchange) or zero velocity of the membrane at the edge of the domain (i.e. fixed projected membrane area and no membrane exchange), as described in Methods, Theoretical Modeling, Section 8. For water exchange, we considered adhesion potentials with different strengths and a fixed pressure difference ensemble. Soft potentials allow for changes in the distance between the adhered part of the membrane and the substrate, and hence for enclosed volume exchange, as does the fixed pressure ensemble, whereas protrusion volume was nearly fixed by considering very stiff adhesion potentials. We found that although the results quantitatively depended somewhat on these conditions (Suppl. Figs. 7 and 9c), the overall mechanisms were robust across conditions. The only qualitative exception was the case of fixed pressure for tubes, where tubes uniformly elongated without exhibiting pearling followed by the tube-bud coexistence, Suppl. Fig. 8, showing that some degree of volume confinement is required for the experimentally observed sequence of events of tube reshaping. We also found that if volume confinement was too large, the bud did not significantly shrink during neck elongation as the volume inside the small tube is very small. This is now discussed in Section 8.3 of Methods, Theoretical Modeling.

23. On page 17, the authors write “because the later is changing” and it should be “latter”

We have now substantially re-written the Theoretical model methods and this error has been erased.

24. It would be valuable to cite this review article in the proper place or places: M. Simunovic, G. A. Voth, A. Callan-Jones, and P. Bassereau, “When Physics Takes Over: BAR Proteins and Membrane Curvature”, *Trends Cell Biol.* **25**, 780-792 (2015).

It is indeed an important manuscript wrapping up the descriptions on how BAR proteins have a mechanochemical role when sensing and reshaping membranes. We now reference it in the introduction of the manuscript.

Comments about figures:

25. In Fig. 2b-e, the energy density landscapes are “according to our mean field density functional theory”. Is it correct to assume these energy density landscapes were calculated using Eq. 11 and not the explicit parameterization given by Eq. 12,15? This should be clarified in caption and in text.

Yes, the referee is correct in assuming that these plots were generated using Eq. 11 (12 in the revised manuscript). This has been clarified in the caption and in Section 4.2 of Methods, Theoretical Modeling. In any case, as shown in Suppl. Fig. 3 both models agree very well.

26. In the caption of Fig. 2b-e, the plots are described as energy density landscapes. However, these are free energies and should be described in agreement with Suppl. Fig. 3 and the in-text description on page 15.

In these panels, we represent the areal free energy density, that is the free energy divided by the membrane area. Since these plots are for planar, cylindrical or spherical surfaces, which have uniform curvature, this quantity allows us to compare different structures. We have changed all labels from “Energy density” to “Free-energy density”. The definition of “the areal free-energy density” has been introduced in Methods, Theoretical Modeling after Eq. (12).

27. In Fig. 2b-e, are the depictions of the isotropic and nematic phases to scale (i.e., is the protein length proportional to the membrane size)?

In the original submission, our depictions were diagrammatic and did not accurately describe the actual system. Following the referee’s comments, in the revised manuscript we have generated microscopic realizations of molecular organization consistent with coverage and orientational order of the mean field theory using a Monte Carlo algorithm. These new depictions are introduced in Figure 2 and Suppl. Fig S3. They indeed provide much better physical insight.

28. In Fig. 2b-e, it is unclear how the model accounts for the “crowding effects” that render surface coverage greater than 0.7 impossible. It is unclear from figure caption and theory section.

Our mean field model, described in detail in Tozzi, C., et al, and Arroyo, M., *Soft Matter*, 2021, 10.1039/D0SM01733G, computes the entropy of elongated particles by estimating the excluded area of pairs of particles depending on their orientation with a strict non-overlapping condition. This point has been further clarified in Methods, Theoretical Modeling. A maximum coverage of 0.7 reflects the fact that at maximum packing, the ordered ellipses do not cover completely the membrane surface.

29. In Fig. 2d,e, the proteins in state i are shown dispersed and aggregated, respectively. The proteins in states ii and iii are shown distinctly different ordering with respect to each other (offset or in a line) and with respect to the long axis of the tube (perpendicular or

tilted). It is unclear what aspects of these depictions (and those in Suppl. Fig. 3) are based on model data or hypotheses and could be more misleading than helpful.

Thanks to the referee's comment, we have improved these depictions as discussed in 27 above. We have generated microscopic realizations of molecular organization consistent with coverage and orientational order of the mean field theory using a Monte Carlo algorithm. These new depictions are introduced in Figure 2 and Suppl. Fig S3.

30. In Fig. 2f, there is text that is cutoff. "Chemical distributio"

This has been corrected in the figure (now Fig. 2h)

31. Fig. 3a compares simulation and experiments showing bud reshaping. However, the simulation does not have a timescale associated with it.

Thanks to the referee's suggestion about the dynamics of bulk diffusion, see points 9, 10 and 20 above, we understand now that the snapshots in our experiments can be understood as quasi-static mechano-chemical equilibria at fixed local concentration of dissolved proteins in the immediate vicinity of the SLB, which slowly varies. According to this new understanding, we have significantly changed the way we interpret our experiments and simulations. We first kept the simulations without timescales to qualitatively compare them with reshaping observed experimentally, and we subsequently address the difference in timescale.

32. Fig. 3f is a model (left) and experimental (right) timeseries of tube reshaping.

- a) However, the end points of the tube reshaping are qualitatively different. The model has pearls remaining while the experimental system does not appear to. It is unclear why this occurs. Does the model overstabilize the pearl? Is the effective protein coverage lower compared to the experiment?
- b) Fig. 3f left does not have any estimate of time. What is the model time of the series of snapshots?
 - a. The precise shape of reshaped tubes at high protein chemical potential depends somewhat on the simulation details, e.g. on the mechanical ensemble, that we have now explored in detail in Suppl. Fig. S7. Our simulations seem to over-stabilize a pearl at the base of the protrusion, but the main point of the simulations is the co-existence of pearls and thin tubes, with the progressive reduction in the number of pearls.
 - b. We kept the snapshot as they are without timescale as just explained in our response to the previous question # 31.

33. In Fig. 3j, the concentrations should be in μM for consistency.

We have corrected the plot, now Fig 3f., using the μm unit instead of nm.

34. In Fig. 3j right, experimental images of pressurized caps are shown. An explanation of how these images differ from the bud structures previously shown would be helpful.

We apologize for not sufficiently clarifying the difference between both types of structures. Buds are obtained from a cycle of stretch and compression of the bilayer, without any changes in buffer composition. Therefore, buds are formed upon lipid bilayer compression. On the contrary, caps are obtained not by applying any stretch, but by submitting the bilayer to a hypo-osmotic shock. This is achieved by exposing the top of the bilayer to a buffer of lower ionic strength. The caps formed adopt a spherical shape and are pressurized, unlike the buds. The buds are connected to the bilayer through a thin neck, while the caps have a large base connecting it to the bilayer, which is why we called them caps. This has been established in detail in a previous publication from two of the co-authors (Staykova, et al, and Stones, Phys Review Letters, 2013, 10.1103/PhysRevLett.110.028101).

We use the pressurized caps to verify that reshaping should be limited by the high pressure. Indeed, no reshaping of the caps was observed even injecting protein at high concentration. By further increasing the protein concentrations, rather than reshaping, we observe that caps cannot withstand the extra pressure due to protein binding, which results in tube lysis. We now clarified the text to better explain this. We also previously introduced more clearly the different shapes, with different properties, that we work with and that had been described in Staykova, el al, and Stones, Phys. Review Letters, 2013, 10.1103/PhysRevLett.110.028101.

35. The caption of Suppl. Fig. 2c states “but no reshaping in the form of thin tubes is observed.” However, there is no estimation of the tube radius. How do the authors determine that the tubes are not thin nor become thin and what is the definition of thin used here?

The reviewer correctly notes that we did not provide a sufficiently clear explanation of what occurs to the tubes when no protein is injected. In fact, we cannot directly say that no reshaping occurs in this case. A fraction of tubes spontaneously detach as their connection with the bilayer goes through a thin neck, which can easily rupture. The tubes that do not detach progressively shorten, widen, and transform into a structure with a spherical/bud shape, until at some point they get immobilized on top of the bilayer. We have quantified in the control experiment the different times at which these events occur, which is generally a very long time (from 10 to 20 minutes). In the different controls that we have performed in presence of Neutravidin, we observed the same events, either no change in the shape of the moving tube, or rounding of the tube. This relaxation process is thus opposite to the pearling and elongation of tubes happening in presence of amphiphysin. When we stated “no reshaping in the form of thin tubes is observed” we meant that we were not observing this progressive elongation of tubes, but we agree that tube diameter itself could not be measured and thus the statement could be confusing. We now clarified these phenomena in the text in to clearly differentiate the tube relaxation mechanism occurring to the formed free-standing tubes as time passes, with the reshaping due to Amphiphysin action [section A novel system to study the dynamics of curved membrane remodeling by BAR proteins].

36. In the caption of Suppl. Fig. 3, describes the landscapes as “landscape of the free-energy density” instead of “energy density landscape”. Is it correct to assume these are interchangeable terms?

Yes. This has been clarified in the revised manuscript, see also our reply to point 26.

37. Suppl. Fig. 4a shows a timeseries of vesicle consumption. When comparing the final structure of Suppl. Fig. 4a. to structures in the phase diagram of Suppl. Fig. 4b, it appears that the final structure is not the equilibrated structure because the bud in the final panel is a tear-drop shape rather than nearly-spherical. It would helpful to add to this timeseries as the final relaxation occurs.

In the new Suppl. Fig. 4b we show the dynamics up to a later point at which the bud has almost completely disappeared. The time evolution of the tube length suggests that beyond a certain point, the tube-bud system becomes unstable and the dynamical process of tube elongation and vesicle consumption becomes very fast. There is some anecdotal experimental evidence of this. We are currently examining theoretically this phenomenon in detail.

38. Suppl. Fig. 5a. “Mean Intensity Protein = $RI_{lipid} / A_{protein}$ ” What is RI_{lipid} and how does it differ from $RI_{bilayer}$?

We apologize for this mistake and we corrected the figure by replacing RI_{lipid} with $RI_{protein}$.

39. Suppl. Fig. 6d shows a timeseries without units of time and shows an end state of many disconnected pearls not seen in Suppl. Fig. 6c. It is unclear that this is a stable state or an intermediate toward an elongated tube. The caption also omits the bulk concentration of amphiphysin and the reader cannot compare the unlabeled control to the labeled experiments. Overall, the similarity between unlabeled control and labelled experiments compare.

In the snapshots shown in Suppl. Fig. 6c, now Suppl. Fig. 4c, the lipid tube is sometimes difficult to see due to its much dimmer fluorescence. To show that the tube is however present, we now added the movies of the corresponding experiment in which the dynamics enables to better visualize the overall shape of the reshaped tube (suppl. Video S16). We also added a couple of snapshot series from another set of tubes from the control experiment (Suppl. Fig. S4c). We also apologize for not having specified the bulk concentration (300 nM) and we now mention it in the figure legend.

40. In Suppl. Fig. 7b, what is the difference between the variations of orange and green colors? Is there a significance to the size of the markers in the left and right of Suppl. Fig.7b?

In the new Suppl. Fig 9b, the variations between orange and green colors corresponds to different curved templates reshaped at the same bulk protein concentration as mentioned in the legend. There is no significance to the size of the markers, we apologize for this imprecision and we have now corrected it in the graph of Suppl. Fig. 9b.

Reviewer #2 (Remarks to the Author):

The manuscript by Le Roux et al. and Roca-Cusachs examines membrane shape changes induced by elongated curved proteins, commonly known as BAR domain proteins. The authors use lipid bilayer membranes deposited onto activated (oxidized) PDMS surfaces that can be stretched so as to increase area available for liposomes to fuse. This membrane excess area can be consumed by the formation of membrane tubules upon stress relaxation in the PDMS sheet. The authors study interactions of the BAR protein amphiphysin with this system.

In a second thrust, the authors developed a theory that allows to study disorder-nematic ordering transitions of curved proteins on membrane substrates. The authors find interesting couplings between the shape of the membrane and phase diagrams that separate nematic from disordered states as a function of density of protein on the membrane, and membrane shape.

In a third thrust, the authors put cells onto their substrates and find similar curvature transitions at the plasma membrane as those in the model membrane system. While the overall scope of the study is impressive, **I am not convinced that the experimental results significantly add to the existing literature, which is barely discussed in the context of the authors' findings.** The experimental results appear to be somewhat **disconnected** from the theory insight. For example, **experiments do not analyze nematic ordering of proteins on the membrane, a key prediction of the calculations.**

We thank the reviewer for his comments and questions, and we would like to clarify why we are convinced that our experimental results are a significant addition to the existing literature. Previous studies have mainly worked at equilibrium, for instance by experimentally and theoretically studying tube pulling (Sorre et al, and Bassereau, PNAS, 2012, 10.1073/pnas.1103594108, C. Prévost, et al, and M Simunovic, JoVE, 2017, 10.3791/56086, C. Prevost, et al, and Bassereau, Nat. Com., 2015, 10.1038/ncomms9529), SLIC (V.K. Bhatia et al, and D. Stamou, EMBO J., 2009, 10.1038/emboj.2009.261) or wavy bilayers (W-T Hsieh, et al, and T. Baumgart, Langmuir, 2012, 10.1021/la302205b) assays. These studies showed how sensing and reshaping depend on the geometry of the lipid template, its tension, and protein density. Indeed, it was demonstrated that at already low concentrations, BAR protein sorted differently depending on the previous parameters listed. Then at high concentration, the protein is able to reshape the membrane. Reshaping has been mainly studied in the tube pulling assay combined with fluorescence microscopy, where the tube is constricted to the preferred diameter of the protein, or in the liposome tubulation assay combined with electron microscopy (EM), where proteins tubulate liposomes (J. Adam et al, and N. Mizuno, Sci Rep, 2015, 10.1038/srep15452, C. Mim et al, and V M Unger, Cell, 2012, 10.1016/j.cell.2012.01.048, A Frost, et al, and Vinzenz M. Unger, Cell, 2008, 10.1016/j.cell.2007.12.041). In these works, reconstructed images have shown the nematic organization of the scaffold formed by the protein at an atomic resolution. Coarse grain simulations have computed the initial self-assembly of the proteins, as a prelude for membrane remodeling (Simunovic, et al, and Voth, PNAS, 2013, 10.1073/pnas.1309819110).

However, a dynamical and mechanistic description of the process, showing the evolution from a completely random distribution of protein in a low-curvature membrane, to a completely formed scaffold and nematic state everywhere in the template, was lacking, both experimentally and theoretically. Our results show this process, demonstrating how the reshaping mechanism proceeds by the progressive transition between two co-existing states characterized in terms of shape (low isotropic curvature to high anisotropic curvature) and protein organization (low-coverage isotropic to high-coverage nematic). In intermediate states, the system is characterized by a mixture of these two states. Our work shows how the geometry influences the establishment of the mixture. For buds, the neck is a template for the high-curvature state. Interestingly, for thick tubes generated by mechanical stretch, these two states are nucleated through a pearling transition. Such pearling transition resulting from isotropic spontaneous curvature had been identified before, but never as a precursor of a transition between a low-curvature-low-coverage-isotropic state to a high-curvature-high-coverage-nematic state. During reshaping, the mechanisms of curvature sensing and generation become intertwined. BAR proteins may need a sufficient concentration to start reshaping significantly the lipid template, but if they modify this template, this modification will increase their local density due to curvature sensing. This positive feedback loop had not been visualized in such a dynamic manner. Our results also show how remodeling of templates can occur at moderate coverage before a nematic state is reached, as shown by pearled tubes. Importantly, our experiments also imply that the Amphiphysin BAR protein can respond to mechanical force, since bilayer compression is the signal leading to the formation of curved templates that are then remodeled by Amphiphysin. This is what we further investigate in cells, finding that cell compression triggers Amphiphysin tubulation. This evidences that Amphiphysin indeed responds to a mechanical signal, opening the door to possible mechanotransduction mechanisms initiated by amphiphysin or by BAR proteins in general.

The project therefore seems to be preliminary in as far as the junction between theory and experiments is concerned. I have the following more specific comments for the authors to consider:

Regarding the connection between theoretical predictions and experimental results, we respectfully disagree with the reviewer's opinion that they are disconnected. The main prediction of the model is the dynamic evolution of template shape, which we systematically analyzed in several conditions. Further, we also experimentally test predictions on protein coverage, and the relative protein concentrations in nematic and isotropic phases. The only prediction that we do not study experimentally is nematic orientation, as the reviewer points out. However, we note that this would be extremely challenging, particularly considering the dynamic nature of our experiments. Measurement of protein orientation has recently been achieved (V. Swaminathan, et al, and Waterman, PNAS, 2017, 10.1073/pnas.1701136114), but in integrins in focal adhesions, which move and reorient very slowly. In our fast moving structures this would be impossible. We tried to perform bilayer fixations after the protein has bound, with the aim of collecting electron microscopy images, but we were not successful. Given this experimental barrier, we think it is reasonable to assume that once reshaping starts, spherical/tubular curvatures are likely to be associated with isotropic/nematic orientation. We now introduced more extensively the existing literature in the introduction, and explained how our work is different from previous studies and adds novelty to the field.

Before answering the subsequent questions, we want to already mention that we have made substantial additions to the model and broadened the range of cases simulated, in order to achieve a better junction between the theoretical and the experimental parts. In particular, we have considered a diversity of protrusion shapes and of mechanical ensembles (controlling the ability of a protrusion to exchange membrane area or enclosed fluid volume with its surroundings) to establish the generality and robustness of the reshaping mechanisms identified here. Importantly, we have identified that diffusion in the bulk medium following protein injection is the slowest process, which provides a better characterization of the experimental dynamics and a match with the model in terms of timescales.

Major:

1) The authors find in their theory, that for a wide range of bud diameters and protein concentrations, the enrichment on tubular regions versus bud regions was approximately two-fold. This result is confirmed by experiments. **I believe that this result may contradict early studies on the curvature sorting of proteins on membranes that are available in the literature. The result also appears to contradict a recent study by Larsen et al. and Stamou, ACS Central Science 2020, which is not cited.**

Regarding comparison to the literature, perhaps one of the most directly related studies is that by Sorre et al, and Bassereau (PNAS, 2012, 10.1073/pnas.1103594108), where they computed the relative enrichment between the GUVs which are very large (in the range of 5-10 μm diameter) and very thin tubes (in the range of 50-150 nm diameter). Enrichment on the tube in their work has been quantified in the “high density” protein concentration regime, as it is the case in our experiments. The summary table from their supplementary material (table S1) indicates relative enrichments ranging from 1.8 to 5.4, with quite large errors in the estimation of the coverage as this quantification is very challenging. Our findings (2-fold enrichment) are thus consistent with the lower end of their results, potentially because the vesicles we are considering are 5 to 10-fold smaller than theirs (and thus with higher curvature, reducing the relative enrichment between the tube and the vesicle). Additionally, we note that differences in membrane tension, and also lipid or buffer composition, may have an influence. To clarify this, we now compare our findings to those of Sorre et al. in the manuscript [end of section Physical parameters governing the reshaping of mechanically-induced curved lipid template].

The work from Larsen et al. and Stamou, ACS Central Science 2020 mentioned by the reviewer uses a very different system than ours. It explores the differences between several curvature sensing proteins with sensing properties driven by a membrane inserting motif, in absence of the BAR domain. They use the SLIC assay, at a low protein concentration corresponding to the sensing regime, where no reshaping is observed, and compare vesicles with diameters between 20 and 400 nm with nanotubes spanning a similar range of diameters. They obtained a very high enrichment on very small vesicles (less than 200 nm diameter), and much lower on nanotubes with similar diameters. Their findings point out that gaussian curvature, and not mean curvature, is the driving factor for the observed higher sorting on a spherical shape compared with a cylindrical shape. They explore how the hydrophobicity of the anchor and its shape influences the lateral pressure in the bilayer and drives the differences between the geometries.

We thank the referee for raising this reference, that we cite in the revised manuscript. It is a very nice and interesting work, and we acknowledge that in the case of N- BAR proteins having both an alpha helix and a BAR domain, both domains certainly work in synergy and regulate the sensing and reshaping properties of the protein, see Section 6 Methods, Theoretical Modeling in the revised manuscript. However, we note that the experimental conditions described in the previous paragraph and the scope of the study from Stamou's publication are very different from ours and cannot be directly compared. They examine only protein sorting at very low bulk concentration, where no reshaping occurs, on highly curved lipid templates very different from ours. We are not considering this very low concentration regime where only sensing occurs and start with free-standing tubes and buds of moderate curvatures, where lipid packing defects are not favored. Furthermore, we do not promote lipid packing defects due to composition, as we use lipids that have a cylindrical shape (we do not use the conical-shape DOPE lipid). In Stamou's work, the importance of the pressure profile is highlighted as it is a driving parameter for anchor insertion. In their in-vitro set up, liposomes and tubes stand on their own on top of the substrate whereas in our system, the tubes and buds are connected to the lipid bilayer. In the supplementary material of a previous publication from two co-workers of our manuscript (Staykova, et al, and Stone, Phys Review Letters, 2013, 10.1103/PhysRevLett.110.028101), a model of such templates connected to the lipid patch was developed. A key result is that the adhered membrane continuously connected to the protrusions enables the relaxation of bilayer packing asymmetries despite the high curvature. In a closed system, such as the vesicles in Stamou's work, the packing and stress asymmetry between monolayers is essentially frozen given the slow rate of flip-flop and leads to packing defects in the outer leaflet. This is not expected in our system, and this being the main mechanism explaining their results, it may not be pertinent here. With all this, we think that our results cannot be directly compared with the mentioned publication by Stamou. We now clarified the scope of our study and discussed the relative roles of both domains in reshaping curved templates in our Methods, Theoretical Model.

2) The bilayers formed on the PDMS substrates look highly heterogeneous, suggesting that a complex curvature pattern may exist in those-substrate supported bilayers. It is not clear how that complication would affect the comparison between theory and experiments. Furthermore, solid supported lipid bilayers may experience asymmetry regarding their trans-bilayer stress distribution, due to interactions with the substrate. For example, it has been shown that solid supported membranes develop asymmetric lipid compositions. How would such effects factor into experimental observations as well as comparison between theory and experiments?

We agree with the reviewer that the bilayer's fluorescence is not completely homogeneous. This is largely because vesicles are present on top of the monolayer, particularly before stretch. During stretch, an important fraction of these is absorbed into the monolayer, preventing rupture (see more detailed description in Staykova et al. and Stone, Phys. Review Letters, 2013, 10.1103/PhysRevLett.110.028101). Additionally, we adjust the plasma treatment to ensure sufficient friction with the substrate (necessary step as highlighted in Stubbington et al, and Staykova, Soft Matter, 2017, 10.1039/C6SM00786D), and this process can lead to local inhomogeneities.

To verify whether inhomogeneities in the monolayer itself could lead to mechanical heterogeneity, we quantified FRAP experiments at the right and left sides of bleached lines. We verified that the recovery was symmetric (Suppl. Fig. S1c). Still, we

acknowledge there may be nanometer-scale deformations of the PDMS affecting membrane curvature. However, we note that our study focuses on the reshaping of membrane templates (tubes and buds) generated by membrane compression. These structures are largely free-standing, and are only connected to the substrate at one end. In both experiments and model, the flat membrane bilayer surrounding templates thus acts simply as a “reservoir” of membrane area, with a certain friction with the substrate, and as a reservoir of enclosed interstitial fluid. Of course, Amphiphysin may interact not only with templates but also with nanometer-scale deformations on the flat membrane bilayer surrounding templates, but such interactions do not lead to membrane tubulation (see control experiments in unstretched membranes, fig. S2e-f), and were not the subject of our work.

The heterogeneity of shape of protrusions can be seen as means to probe the diverse templates that may exist in cells where proteins may act. In the theoretical model, we have addressed this variability by considering protrusions, spherical and tubular, of different geometry. Furthermore, since the way protrusions exchange membrane area and enclosed fluid volume with the surrounding reservoir is unknown, we have now tested the robustness of the proposed mechanisms by considering different mechanical ensembles, Section 8 in Methods, Theoretical Model in the revised manuscript. Our results are very robust in showing two prototypical paths from low-curvature structures to high-curvature and high coverage tubes: (1) elongation of high curvature, high coverage and nematically arranged necks connected to spherical buds and (2) pearling of non-nematic tubes, leading to the creation of necks and to multiple copies of mechanism (1). (You can also consult the answer to point 15 of Referee #1.)

Regarding asymmetry between leaflets, we agree that it may occur in our set up as in any in-vitro or cellular system, particularly in the flat part of the bilayer connected to the substrate. However, as stated above our study focus on free-standing structures detached from the substrate, where this effect should not be relevant. Further, it has been shown that the lipid chain, and not the head, is the driving factor of asymmetry (D.C. Ling, et al, and Longo, *Biophys J.* 2006, 10.1529/biophysj.105.067066). Here we used only one type of chain (dioleoyl lipid chains), and therefore this should not lead to asymmetry. We now specified this in the Methods part.

3) The authors mention that they did not control the dynamics of protein delivery to their membranes. However, it could be argued that microfluidics has become sufficiently ubiquitous to claim that the experimental design used here may not be state-of-the art. It seems to me that any experiment describing dynamic aspects of protein-lipid interactions ought to carefully control the dynamics of protein delivery in order to be able to produce meaningful and reproducible data.

Before entering the topic of protein delivery control, we want to insist on the fact that we designed an advanced and completely novel setup. To our knowledge, there is no literature on supported lipid bilayer stretch, except from the setup developed by one of the authors (Staykova, M., et al, and Stone, *Phys. Review Letters*, 2013, 10.1103/PhysRevLett.110.028101). We took inspiration from this setup and completely redesigned it to implement a controllable system. We optimized the plasma cleaning parameters to render supported lipid bilayers of very specific properties, as these must mimic the behavior of a cellular plasma membrane under compression. In fact, very few in-vitro lipid bilayer systems can model a cell membrane in the form of a fluid lipid

bilayer characterized by both “lipid reserves” (J. Steinkülher, et al, and R. Dimova, *BioRxiv*, 2020, 10.1101/2020.07.13.198333v1) and friction with the substrate. The necessity of forming a charged bilayer, essential for Amphiphysin binding, made this more difficult and demanded experimental tuning of the set up. As we wanted to stretch the bilayer, we used a thin PDMS membrane, and we designed a protocol to obtain a patterned bilayer. This created a specific pattern upon stretch or compression, enabling us to distinguish specific areas of the bilayer, and to measure applied stretch by comparing stretched and unstretched patterns. We also designed the protocol such that the patterned bilayer remains confined inside of a small, bonded ring, thin enough not to impede bilayer stretching, and to keep as an outcome an equibiaxial homogeneous strain. We controlled stretch through a controlling card and Labview-based software. This allowed us to stretch slowly enough so that liposomes would incorporate and avoid bilayer rupture, and to compress slowly enough to follow tube and bud formation in real time (having controlled repeatedly that immediate compression also led to such curved template formation). We combined this device with a spinning disk confocal microscope. This confocal set up ensures a good signal of the fluorescent protein bound to the bilayer as we cannot rinse the protein from the bulk, but also enables us to acquire very rapid frames in two fluorescent channels, of a sufficiently large image, at high resolution (60* magnification objective). The use of the objective itself has been optimized as the sliding of the PDMS membrane upon stretch complicates the contact between the objective and the substrate. The camera for imaging also needs to be a very fast and high-quality equipment, and combined with the software used, it enables a real-time display of the acquired snapshots. Indeed, the PDMS membrane quickly goes out of focus and needs to be manually adjusted in real time, as the focusing plane is not the image of highest intensity. Altogether, we believe we have set a novel and state-of-the-art device.

Our statement about uncontrolled delivery was meant to explain that we did not apply a given Amphiphysin concentration instantaneously, but that concentration increased gradually due to diffusion. We apologize since our statement incorrectly suggested that our protocol was not controlled or reproducible, which was not at all the case. Indeed, we controlled protein delivery by always injecting it in the same manner, adding the same volume of the protein solution in the same initial buffer volume. This of course implies that there is diffusion, but this is a limitation even in microfluidic systems, especially if the binding affinity of the protein to the ligand is high. The way to improve the delivery is to use high flow rates. To this end, we considered imposing a flow once the curved templates were formed, as it would improve protein delivery and permit better imaging of tubes (that would align with flow). Unfortunately, we quickly noticed that tubes were extremely sensitive to any perturbation, and could detach very easily, as we now mention in the text and as can be visualized in Suppl. Video S3). We have quantified the percentage of tubes spontaneously detaching after the release of the stretch, in which neither a flow nor an injection were imposed. More than half detached, several doing it directly at the end of the complete bilayer release, others detached randomly during the next 5 minutes. Even when membranes were accidentally shaken during experiments, tubes quickly detached, so imposing flow was out of the question. Another option would be to use a diffusion chamber (as in Mally, M., et al, and Derganc, *RCS Advances*, 2017, 10.1039/C7RA05584F, for instance), but this would also imply equilibration times of the order of minutes. Another difficulty in our system is that building a chamber is not straightforward as we need to plasma clean the substrate and immediately apply the liposome solution, on top of a membrane that needs to be stretched. This would be very complex inside of a closed chamber as a chamber attached to the top of the bilayer would

severely disrupt stretch magnitude and symmetry. We want to stress that the literature related to membranes reshaping by BAR proteins almost completely focuses on the equilibrium state and as such, most set up use a microinjection set up implying equilibration times larger than minutes (Sorre et al, and Bassereau, PNAS, 2012, 10.1073/pnas.1103594108). Even a very recent single molecule study uses such a set up without microfluidic systems (Bashkirov, P., et al, and Frolov, Nature Protocols, 2020), although they are working inside of a chamber.

Despite these limitations, as we mentioned above we controlled protein delivery by always injecting it in the same manner, adding the same volume of the protein solution in the same initial buffer volume. Since our delivery was systematic and following the comments of this referee and of referee #1, in the revised manuscript we modeled protein delivery through bulk diffusion in the medium from the injection point to the close vicinity of the SLB. This analysis showed that diffusion in the bulk is the slowest process as compared to adsorption, membrane diffusion and membrane reshaping, enabling us to interpret our observations as quasi-steady states at a given dissolved protein chemical potential (Suppl. Fig S5 and S6). This has affected the way we present our data in the revised manuscript.

4) The cellular experiments aim to show that the imposition of membrane curvature can lead to the triggering of BAR protein responses. Similar coupling between imposed curvature and intra-cellular proteins has previously been described, e.g. by Galic et al. and Meyer, Nat Cell Bio 2012. It is not clear to what extent the cellular experiments shown here move significantly beyond the existing literature.

In the cited paper, the authors found that externally “pushing” the membrane by nanocones, or internally “pulling” it via actin polymerization and fiber formation, triggered the recruitment of N-BAR proteins (Galic et al. and Meyer, Nat Cell Bio 2012, 10.1038/ncb2533). This work is highly relevant since cells may be located on top of rough substrates in the body, and actin polymerization certainly pulls the membrane inward in several physiological situations. In a later publication, M. Galic, et al, and Meyer, eLife, 2014, 10.7554/eLife.03116, examined how an N-BAR protein is also recruited at actin polymerization sites in neurons, presumably through curvature sensing, and how it reduces local actin polymerization due to the Rac GAP activity of that N-BAR. Another work from Tsujita et al, Itoh, Nat. Cell. Biol. 2015, 10.1038/ncb3162, found how an F-BAR was binding to membrane invaginations at the leading edge and further recruited N-WASP, favoring actin polymerization stabilizing the protrusions. In all these publications, membrane deformations are due to imposed curvature, which creates tensed curved structures at the plasma membrane. In contrast, in our set up we do not impose any curvature: we merely apply cell stretch followed by compression, which is another very frequent phenomenon physiologically. This mechanical signal triggers the formation of free-standing tubes or buds, which in turn trigger interactions with BAR proteins. Previous work in our group has shown that these curved templates are devoid of actin fibers (Komalska, et al, and Roca-Cusachs, Nat. Comm., 2015, 10.1038/ncomms8292). Thus, our study is dynamical and is not induced by specific topography but by very generic stimulus, bilayer stretch and compression. The fundamental novelty of our finding is that we demonstrate a new mechanosensing mechanism, by which a cell stretch cycle can trigger a membrane reshaping event. In this mechanism, membrane curvature is not the originating signal, but an intermediate step between the mechanical signal (stretch) and the subsequent membrane response. Another important difference is that

these works all focused on local enrichment (therefore, in the sensing regime of the protein) of BAR proteins, but not in membrane deformation induced by the proteins. In our work, we witness a different kind of protein response: as a result of membrane mechanical compression, Amphiphysin directly remodels the membrane itself, as we can attribute membrane tubulation to the action of Amphiphysin. We are not aware of other works having highlighted such finding. We have now emphasized this in the introduction and discussion parts of the manuscript.

Minor:

1) The authors use a somewhat unusual lipid composition that considers the headgroups PC, PS, and PA. I am not aware that PA is present in the plasma membrane to significant amounts. Why not work with a physiologically more relevant lipid mixture?

The challenge of our in-vitro system is to replicate, to the extent possible, the cellular membrane response upon compression. For this, we need adhesion to the substrate and “lipid” reserves (in the cells present in form of ruffles, caveolae, exocytosis mechanisms, in our system as liposomes on top of the bilayer), but at the same time we need to create a fluid bilayer. We achieved this by tuning the plasma cleaning time, enabling us to reproduce the plasma membrane response to compression (extensively studied in Komalska, et al, and Roca-Cusachs, Nat. Com., 2015, 10.1038/ncomms8292). To further promote Amphiphysin binding to the bilayer lipid charges were necessary, as BAR protein binding to the bilayer is initially driven by electrostatic interactions with the lipids. However, this complicated the obtention of a fluid bilayer. We chose as a basis a mixture of DOPC and DOPS, major building blocks of the internal leaflet of the plasma membrane, but we failed in using DOPE (used in one of the works we used as a reference, Sorre et al, PNAS, and Bassereau, 2012, 10.1073/pnas.1103594108) as we could not obtain a fluid bilayer in the presence of such lipid. The use of phosphoinositides is also very delicate, as their handling is very sensitive and as they may have complicated the formation of the desired bilayer. We decided to add DOPA to further promote Amphiphysin binding, as it was shown to specifically recruit amphiphysin in Takei, et al, and De Camilli, Cell, 1998, 10.1016/S0092-8674(00)81228-3. DOPA enriched the bilayer with additional negative charges as DOPS; interestingly, it is a physiological lipid in the inner leaflet of the plasma membrane (Nishioka et al., and Kiyokawa, J. Biol Cell. 2010, 10.1074/jbc.M110.153007) also well known to have signaling roles in cells (Wang, et al, and Zhang, Progress in Lipid Research, 2006, 10.1016/j.plipres.2006.01.005). We have now clarified this in the Methods part of the manuscript.

2) The control of adding neutravidin, which is not observed to bind to the bilayer, would have been more useful if done with a protein that does bind, but that is not supposed to generate membrane curvature. What is the addition of a non-binding molecule supposed to test? It does not seem that this control was motivated, or the results put into context.

This control intended to test whether having a high protein concentration in the bulk, by itself, could influence reshaping of the lipid templates in some way. This was suggested by a researcher from the field, but we agree that the control of a protein binding to the bilayer without reshaping is also highly relevant. The choice of such protein is not straightforward as many proteins insert in the bilayer in order to bind to it and may also deform it. Therefore, we decided to use the Neutravidin protein as well, in combination with a small amount of biotin in the bilayer. As expected, this did not lead to a strong protein signal on the curved templates, since the protein is not specifically enriched there.

Interestingly, some time after injection the tubes started to bind to the bilayer, most likely because neutravidin (with 4 biotin binding sites) crosslinked tubes to the bilayer. This nicely shows that biotin was indeed binding to the bilayer. Importantly, before this event, we could monitor that the tubes and buds behaved as in the control case as well in this experimental set up. We included this additional control in the text [section A novel system to study the dynamics of curved membranes remodeling by BAR proteins], as well as added the corresponding images in Suppl. Fig S2 and the corresponding video (Suppl. Video S5).

3) It is unclear to what extent the authors considered protein crowding effects and their coupling with membrane curvature in their model.

The model in the original submission did account for crowding of BAR domains on the membrane since steric interactions are at the core of the mean field theory. Beyond crowding on the bilayer surface, we acknowledge that disordered domains in Amphiphysin can affect its curvature activity regarding reshaping and sensing, as shown for instance by Stachowiak and collaborators. We added a new Section 6 in Methods, Theoretical Modeling, discussing how different mechanisms coupling protein coverage to membrane curvature are mapped to our model. We argue that the curvature sensing effect of disordered domains through changes in chain entropy is effectively included in our model. However, the crowding effect at high coverage of such disordered domains at a distance from the bilayer surface is in principle not included. To test its effect, we extended our theory to account for crowding of these domains at a distance away from the membrane and found that in the high coverage regime their contribution is small, Methods, Theoretical model and Suppl. Fig. 11.

4) It would be helpful to clearly lay out what parameters of the theory are experimentally known, and which ones are unknown.

To address this point, we have added a table in Methods, Theoretical Modeling, where references substantiating our choices are given. For some parameters, there is no available data. We discuss our choices in Section 5 of this supplement.

Reviewer #3 (Remarks to the Author):

Review of “Dynamic Mechanochemical Feedback Between Curved Membranes and BAR Protein Self-Organization”

Le Roux et al. demonstrate that the BAR domain-containing protein Amphiphysin binds to and induces morphological changes in lipid bilayer nanotubes. This study introduces a new experimental technique with control over membrane tension for producing and imaging supported lipid bilayers. Using this system, the authors first stretch and then relax membranes to generate low-tension and low-curvature membrane “templates.” The authors then monitored Alexa488-Amphiphysin binding to these templates over time. Amphiphysin bound to and stabilized bilayer nanotubes and, at high concentrations, remodeled them into pearled tubes and thin necks. The authors complemented these observations with dynamical models that account for the energetics of membrane bending, protein binding, and isotropic versus nematic protein ordering. Their model predicted that, over time, Amphiphysin would concentrate on high-curvature membrane cylinders versus lower curvature vesicles, leading to pearling. Mechanistically, the prediction is that Amphiphysin-induced membrane remodeling emerges from an isotropic-to-nematic transition of the bound protein molecules. The free energy driving this re-ordering arises from the spontaneous and anisotropic curvatures of Amphiphysin. Experimentally, this prediction may have been verified by measuring the relative intensity of the protein coat on high curvature tubes versus low curvature vesicles. Finally, using the same stretch-and-release technique, the authors demonstrate that analogous membrane tubes form from the plasma membrane of living cells. Like the *in vitro* tubes, plasma membrane tubules appear to recruit over-expressed GFP-Amphiphysin, providing a mechanism for mechanochemical signal transduction in cells.

Overall, this work utilizes a new experimental setup to generate low-tension lipid bilayer tubules and uses this system to explore membrane remodeling dynamics by a well-studied protein. The theoretical model, which is being reviewed in a separate manuscript and provides mostly intuitive results, **may be overly simplistic to offer novel insights**. Additionally, **the concept of an isotropic-to-nematic transition for Amphiphysin is not novel and can be found in the first reference and other sources**.

We thank the reviewer for the constructive comments. Indeed, it is known that at low concentrations/curvatures bar proteins are isotropic and that a high concentrations/curvature they are nematic. We agree with the reviewer that there is ample literature on the architecture of the protein scaffold of BAR proteins around constricted lipid bilayer tubes, as discussed in reference 1 (a review paper on the topic of BAR proteins as molding macromolecules). The nematic arrangement of BAR proteins around reshaped scaffolded tubes has been accurately described by fitting atomic models of BAR dimers into cryo-EM reconstructions of membrane tubules. Other experimental and theoretical studies have shown that a critical concentration of proteins is needed for the proteins to oligomerize and build that scaffold; such studies have mostly described the state of the system at different concentrations, but once the equilibrium is reached. Despite this rich literature, to our knowledge there is only one study looking at the dynamics of the process, but the system under study is a tensed tube that remains in a tubular shape during the entire process (Simunovic and al, and Bassereau, PNAS, 2016, 10.1073/pnas.1606943113). Thus, the dynamics of the isotropic-to-nematic transition, and the intermediate steps involved, which necessarily occur concomitantly to shape

changes, were not known and to our understanding are entirely novel. We have carried out a comprehensive study of these reshaping dynamics, in which we fully describe the nature of the isotropic to nematic transition for curved templates of different sizes and shapes, with mechanical properties and shapes that are usually not addressed in other in-vitro study (size and tension). Our templates are in addition physiologically relevant, as they are also observed in the cellular context. We show how this transition is dependent on the initial shape and mean curvature of the template, and we have witnessed a novel reshaping process of the tubes triggered by a pearling phase occurring at low bulk protein concentration. We have clarified this novelty in several instances of the manuscript.

Regarding the model, we respectfully disagree with the reviewer, and consider that it is not simplistic. We further elaborate this in response to Major concern 3 below. The fact that one part of our model, now published in (Tozzi, et al, *Soft Matter*, 2021, 10.1039/D0SM01733G), provides predictions that agree with intuition is not in our opinion detrimental of its novelty, pertinence and utility to understand the mechanisms of membrane reshaping by Amphiphysin. To our knowledge, this work is the first one to systematically characterize how membrane curvature affects the isotropic to nematic transition for curved and elongated proteins. Of course, it is known that (A) on low-curvature membranes and low coverage BAR proteins adopt an isotropic dilute state whereas (B) at high coverage and after a long incubation they arrange nematically. However, the mechanisms to go from A to B have remained elusive. We show here that they involve heterogeneous and dynamical mixtures of states. To model such mechanisms, in the present manuscript we combine the mean field model in Tozzi, et al, *Soft Matter*, 2021, 10.1039/D0SM01733G valid at fixed membrane shape with a dynamical model for membrane reshaping. The resulting model captures the two-way interplay between membrane shape and protein coverage and orientational organization, and provides a mechanistic explanation for our observations.

Thus, we are convinced that our dynamic characterization, both from an experimental and theoretical perspective, of the non-equilibrium process of reshaping is highly novel.

Overall, we believe that this manuscript requires major and minor revisions before being reconsidered for publication in *Nature Communications*.

Major concerns:

1. The experiments relied on confocal fluorescence microscopy. Unless we are mistaken, the data presented were all single Z slices, rather than Z-stacks or volumes. We are concerned that without sampling in the Z dimension, the quantifications may be systematically in error due to the imaged structures' unknown morphology. For example, a nanotube protruding in the Z direction will appear as a dome or vesicle. Similarly, the morphology of the structures at the end of the bilayer necks cannot be unambiguously identified as spheres from single Z slices. Z stacks should be included in this manuscript to establish the morphologies assigned to the bilayer structures firmly.

We agree with the reviewer that 3D stacks of the structures would be very useful. Unfortunately, and as our images show, the curved structures are free-standing and therefore very “floppy”, their orientation is not stable with time, and they quickly move around their attachment point to the bilayer (as can be seen in the supplementary movies). Tubular sections also wiggle and change shape, as they can easily bend. This implies that

even with the fastest scanning mode of our spinning disk confocal microscope, structures would have significantly changed position and shape between each confocal slice, rendering 3D imaging impossible. The only type of structure that does move slowly are buds before reshaping begins (see for instance Fig. 3a, and supplementary video 10). Even in this case, however, the z-resolution of our confocal (additionally limited by the fact that we are using a deformable thin PDMS membrane as a substrate) is not sufficient to resolve the structure (with sizes in the range of 1 μm) in an informative manner. We even considered using flow to align structures and reduce their movement, but this was not possible either as structures easily detached from the bilayer when perturbed. We also tried to fix the bilayer, in presence or absence of protein, to obtain high spatial resolution of our templates by electron microscopy, but we were not successful in these experiments. Freeze-dry techniques are incompatible with the stretch system set up.

However, we have in fact used this fast movement to our advantage: during our experiments, we carried out fast time-sequence imaging (each 1 s). As the tubes are moving, we can clearly resolve their shape in the experimental movies, and the time frames shown in the figures correspond to those in which structures are as aligned as possible in the XY plane (as opposed to protruding in the Z plane). In the case of buds, we can see in videos that they systematically appear as spherical sections regardless of their orientation. Therefore, we can be confident that the initial shape is that of a bud. In the case of tubes, we now included an additional control in which Neutravidin binds to a biotin-enriched bilayer. After some time, tubes attached to the bilayer, probably because the four neutravidin biotin binding sites promoted tube to bilayer attachment. In this attachment process, when the tube orients parallel to the bilayer, we also can clearly see its cylindrical shape as we now mention in the text and display in Suppl. Fig. 2d and very clearly in Suppl. Video 5.

Regarding quantifications of protein coverage, which is indeed complicated by the 3D nature of the curved templates, we included a geometrical correction, by taking as a reference the fluorescence from the lipid channel (see suppl. Fig. 9a). This enabled us to correct for changes of fluorescence caused by integration of a 3D structure in 2D images. It also corrects for the loss of signal since we are not exactly focusing on the bilayer plane, but a bit above to resolve better the 3D geometry of the moving templates.

We have now stressed these aspects in the results. We are aware that this set up is not as accurate as a tube pulling experiment, which enables an accurate quantification of the signal in a simpler 2D geometry at equilibrium. However, our limitations are inherent to the fact that we are providing a dynamic description of the remodeling event, using templates not sustained by tension.

2. Further, please include more information about the microscopy in the methods section. For example, how are the Z heights for the micrographs in the Figures optimized? This concern is especially critical for quantifying the fluorescence from tubes versus buds and vesicles.

We apologize for not specifying how the z-plane was focused. Due to its deformability (necessary to stretch), the thin PDMS membrane generates vertical drift in the stretch system. As we are interested in the tubes and buds formed on top of the bilayer, we need to focus a bit out of plane, above the bilayer, therefore we cannot automatize the image acquisition as the plane of interest is not the plane of highest intensity. We can also not

choose and fix an initial plane due to the PDMS membrane drift. Therefore, our experimental protocol was to manually focus the image over time during acquisition. As the Micromanager software permits a good visualization of the images as they are acquired, the operator can easily readjust the z-plane to catch as best as possible the curved templates. We are aware that this adds complications to the fluorescent signal quantifications, but as explained above we corrected for geometry by using the fluorescence signal from the membrane channel. We now clarified the focusing strategy in the Methods part of the manuscript.

3. Most of the interpretation of the experimental data relies on a theoretical model. A manuscript describing this model is under review separately. For this review, we assume that the model proves sound. Given the importance of the modeling results to this manuscript, we think this manuscript should have more information about the model and its limitations or that the acceptance of this manuscript should be contingent on the approval of the complete description of the model.

The accompanying manuscript describing the mean field model for the molecular organization of curved proteins on curved surfaces of fixed shape is now accepted and published in *Soft Matter* (Tozzi, et al, *Soft Matter*, 2021, 10.1039/D0SM01733G). This model is based on first principles of statistical mechanics but includes some mean field approximations, which are clearly laid down in this paper. Although the isotropic-to-nematic transition of elongated molecules is well-understood, this is not the case for the role of curvature in this transition. The companion paper addresses this for homogeneous systems with fixed curvature. This model, however, is not enough to study reshaping. For this reason, in the present paper we coupled it with a model for membrane shape dynamics to describe the two-way interplay between shape, coverage and orientational order. To the best of our knowledge, no previous continuum model has accomplished this, although we acknowledge in the main text that coarse-grained discrete models have described related but different phenomena. We believe that this is conveyed more clearly in the revised manuscript.

Pertinent questions include: Does the model contain attractive interactions between protein molecules? If so, are these interactions orientation-dependent? Prior work indicates that BAR domain proteins self-assemble into curvature- and orientation-dependent lattices.

Such interactions can be included in our model but have not been included here. We believe that they should not be very strong for Amphiphysin since otherwise they would lead to protein aggregation, maybe with nematic domains, even on surfaces of small curvature and at low coverage, for which there is no evidence in our experiments. The interactions between proteins are purely steric in our calculations. Orientation-dependent attractive interactions probably play a role in conforming the regular lattice at very high coverage. At the level of our mean-field, such lattice is described in terms of a maximum coverage and an angular probability density function since our model does not include details about the translational order of proteins.

Are interactions between the charged membrane surface and the protein explicitly or implicitly included in the model?

They are implicitly accounted for through U_b , Eq. (9) in the theory supplement, since these interactions support the scaffolding effect of the BAR domain. They are also implicit in the adsorption and desorption rate coefficients.

What are the limitations of using continuum models for highly curved bilayers? When do molecular effects such as induction of asymmetry between the leaflets become significant?

Although the thinner tubes in our experiments have radii only a few-fold larger than the bilayer thickness, our previous work on a related system (Staykova, et al, Phys Review Letters, 2013, 10.1103/PhysRevLett.110.028101) showed that the effect of the bilayer asymmetry was negligible since it can be relaxed through inter-monolayer sliding for a protrusion connected to a SLB acting as a reservoir.

For the purpose of our study, the main limitation of continuum models is molecular specificity, and the difficulty of mapping precise molecular features (such as the scaffolding effect, the partial domain insertion or the effect of bulky disordered domains) to mean-field parameters or functional forms. This difficulty is addressed by our discussion in Section 6 of Methods, Theoretical Model. We should add that a strength of such continuum models is their ability to access the timescales and length scales of our experiments. The mechanisms that we have identified depend on the heterogeneous mixture and dynamical transition between membrane-protein states involving system sizes and time-scales inaccessible to molecular models. Some coarse-grained discrete models may access such transitions with reasonable computational effort but they suffer from comparable limitations regarding molecular specificity.

Overall, our concern is that the model may be overly simplistic for a phenomenon as complicated as membrane remodeling by full-length Amphiphysin.

We hope that the clearer and more extensive discussion of the model (now addressing all the comments of the above comments, in Methods, Theoretical Model) convinces the referee that, confronted with the dichotomy between molecular specificity and realistic length and timescales, our model strikes an interesting balance in that it can access these scales while retaining important aspects of molecular organization such as protein coverage and orientational order, implicitly accounting for molecular mechanisms coupling proteins to curvature, see Section 6 of Methods, Theoretical Model, and solving in a self-consistent way the dynamics of adsorption, membrane diffusion and reshaping. To our knowledge, this is the first continuum model addressing all these aspects, which are involved in the mechanisms of reshaping observed in the experiments and predicted by the model.

4. The authors connect their new experimental setup and model to biology by examining GFP-Amphiphysin binding to plasma membrane tubes in cells and report this as a novel form of mechanosensing and signal transduction—but this idea is not as novel as presented in the manuscript. Contributions to the concept of membrane tension-regulated endocytosis include:

Dai, J., Sheetz, M.P., Wan, X., and Morris, C.E. (1998). Membrane tension in swelling and shrinking molluscan neurons. *J Neurosci* 18, 6681-6692.
Raucher, D., and Sheetz, M.P. (1999). Membrane expansion increases endocytosis rate

during mitosis. *The Journal of Cell Biology* 144, 497-506. Dai, J., Ting-Beall, H.P., and Sheetz, M.P. (1997). The secretion-coupled endocytosis correlates with membrane tension changes in RBL 2H3 cells. *J Gen Physiol* 110, 1-10. Riggi M, Bourgoint C, Macchione M, Matile S, Loewith R, Roux A. (2019) TORC2 controls endocytosis through plasma membrane tension. *J Cell Biol.* 218, 2265-2276.

We agree with the reviewer that many studies, the ones mentioned here and others, have described how plasma membrane tension participates in regulating the balance between endocytosis and exocytosis in cells. These studies coincide in finding a decrease in endocytosis when tension increases. Conversely, stretched cells incorporate lipids from their “reserves” (ruffles, caveolae) or through exocytosis, to avoid an increase in membrane tension, or membrane rupture.

The fundamental difference, and novelty, of our cell experiments is that we are not describing a role of plasma membrane tension, but of plasma membrane shape. Indeed, in our cell experiments we find large differences in amphiphysin tubulation before and after a stretch cycle, even though in both cases the cell is in the same mechanical state (unstretched). Simply, the stretch cycle generates membrane templates, which are then recognized by amphiphysin to reshape the membrane. Importantly, these membrane templates occur naturally due to the stretch cycle, and are not imposed either by cell contractility or nano-scale topography (as studied previously for sensing but not reshaping, see M. Galic, et al, and Meyer, *eLife*, 2014, 10.7554/eLife.03116, and Tsujita et al, Itoh, *Nat. Cell. Biol.* 2015, 10.1038/ncb3162). Thus, we demonstrate a novel mechanosensing mechanism, by which a cell stretch cycle can trigger a membrane reshaping event. In this mechanism, membrane curvature is not the originating signal, but an intermediate step between the mechanical signal (the stretch cycle) and the subsequent membrane response. We have now clarified this and placed it in context in the introduction of the manuscript.

5. Some BAR domain proteins can associate with membranes via more than one surface (Frost et al. 2008), and work has shown that Amphiphysin, in particular, acts through more complex mechanisms than simple induction of spontaneous curvature through its crescent-shaped BAR domain. Physiochemical effects such as amphipathic motif insertion, phase separation, and steric crowding by low complexity domains play increasingly appreciated roles in membrane remodeling in reconstituted systems and cells. See, for example:

Day KJ, Kago G, Wang L, et al. (2019) Liquid-like protein interactions catalyze the assembly of endocytic vesicles. *bioRxiv*.

Snead WT, Zeno WF, Kago G, Perkins RW, Richter JB, Zhao C, Lafer EM, Stachowiak JC. (2018) BAR scaffolds drive membrane fission by crowding disordered domains. *J Cell Biol.* 218, 664-682.

These ideas merit discussion in this manuscript, especially in whether and how they contribute to the theoretical modeling. Please discuss how such effects inform the model of Amphiphysin-induced remodeling within a broader overall discussion of their results concerning the copious literature regarding the action of BAR domain proteins.

As the reviewer correctly points out, Amphiphysin possesses an N-terminal disordered domain folding into an alpha helix when interacting with the lipid bilayer, forming an amphipathic motif capable of inserting (at varying depths) into the bilayer leaflet. Amphiphysin also possesses a long disordered chain coupled to the BAR domain. With respect to N-terminal helices, many N-BAR proteins have comparable “anchors”. The curvature sensing role of such helices has been clearly and extensively described (Antonny, B., *Annu. Rev. Biochem.*, 2011, 10.1146/annurev-biochem-052809-155121, Madsen, et al, and Stamou, *FEBS Letters*, 2010, 10.1016/j.febslet.2010.01.053). Membrane anchoring motifs sense the “lipid packing defects” generated either by specific lipid compositions (DOPE would favor such defect as it has an overall conical shape), highly curved structures, or membrane strain. Curvature sensing driven by the helix requires presence of abundant lipid packing defects that may arise in some locations of the cell, but not in others, a mechanism by which the helix could in fact drive protein localization in cells. Such defects also refine the specific orientation of the BAR domain relative to the bilayer (Gallop, et al, and McMahon, *EMBO J.*, 2006, 10.1038/sj.emboj.7601174). In terms of reshaping, alpha helices have been shown to induce curvature in a few publications (for instance, Peter, et al, and Mc Mahon, *Science*, 2004, 10.1126/science.1092586, found that the N-terminal helix was important but not essential for tubulation), but the scaffolding role of the BAR domain has been much more extensively studied and demonstrated (the scaffold formed by different BAR proteins has been unraveled in reconstruction from electron microscopy images, e.g. in J. Adam et al, and N. Mizuno, *Sci Rep*, 2015, 10.1038/srep15452, C. Mim et al, and V M Unger, *Cell*, 2012, 10.1016/j.cell.2012.01.048, A Frost, et al, and Vinzenz M. Unger, *Cell*, 2008, 10.1016/j.cell.2007.12.041). The relative contribution of both domains in both sensing and reshaping is not fully unraveled and may depend on lipid composition, protein concentration, or template shape and size. For instance, a molecular dynamics simulation carried out to study the respective role of such domains, found that a tight binding of the protein to the bilayer was necessary and ensured by the presence of the anchor, but that the scaffolding role was ensured by the BAR domain. Helices alone could deform the membrane only at very high concentrations. (*Biophysical J.*, Blood et al, and Voth, 2008, 10.1529/biophysj.107.121160). Another more recent study goes in the same direction (Belessiotis-Richards, et al, and Steven, *ACS Nano*, 2020, 10.1021/acsnano.0c05960). Experimental studies have highlighted the importance of the helix for protein anchoring, but not for curvature generation (C. Löw, et al, and Balbach, *Biophys J.*, 2008, 10.1529/biophysj.107.113118, F. Fernandez, et al, and Prieto, *Biophys. J.*, 208, 10.1529/biophysj.107.113118, Z. Chen, et al, and Baumgart, *J. Am. Chem. Soc.*, 2016, 10.1021/jacs.6b06820). F-BAR are not equipped with such helices but still are with no doubt curvature inducers. Absence of such anchoring motif in F-BAR proteins in the form of amphipathic helices would be compensated by an additional lipid binding domain, confirming the dominant role of BAR domains in reshaping membranes.

With respect to disordered domains, Bush, et al, and Stachowiak, *Nat. Com.*, 2015, 10.1038/ncomms8875, found that they participated in membrane bending. In the specific case of Amphiphysin, however, its long disordered region has been found to increase its curvature sensitivity at very low concentration, but to delay reshaping (Zeno, et al, and Stachowiak, *Soft Matter*, 2019, 10.1039/C9SM01495K). Finally, phase separation of BAR proteins is beginning to be described, as in the preprint cited by the reviewer concerning F-BAR proteins, or in work on I-BAR protein (Prévost, et al, *Nat. Com.*, 2015). We are not aware that such phenomenon occurs for Amphiphysin; in fact, we have

checked the possible ability of Amphiphysin to form liquid droplets in solution upon addition of a crowding agent, but we failed to observe any type of phase separation.

To address the role of these different mechanisms in the theoretical model as requested by the reviewer, we have included a new Section 6 in the theory supplement. In short, we argue that, although our model is motivated by the scaffolding effect of the BAR domain, it effectively captures the effect of wedge-like insertions and the curvature sensing low-coverage effect of disordered domains. However, at high coverage, it does not include the crowding effect of such bulky domains. To address this, in the revised manuscript we have adapted a model for the coupling between such domains and curvature in Tozzi, et al, and Arroyo, *New J. Phys.*, 10.1088/1367-2630/ab3ad6 to the present context, finding that the effect is small.

Minor concerns:

1. In the micrographs of amphiphysin-GFP coated tubes, why is the membrane marker visible on membrane caps/vesicles but not the membrane tubes?

In these images, the bud has already been elongated and reshaped, and the fluorescence intensity of the membrane marker itself becomes lower as the tube gets very thin. In general, we can see that the relative signal of the membrane marker on the tube compared with the bud is lower than that of the protein, as the latter becomes enriched on the tube. The reviewer may refer in this question to the control in absence of fluorescence protein. In these snapshots, the tube connecting the pearls is indeed not easily visible, but when we play the video we can clearly see it. We now prepared the corresponding video to make this clearer and added more examples of other tubes from that control experiment, see Suppl. Fig S4 and Suppl. Video S11 and S16.

2. Many references are missing information or improperly formatted.

We apologize for the improper reference formatting and have corrected this.

REVIEWERS' COMMENTS

Reviewer #1 (Remarks to the Author):

Recommendation: Publish after minor revisions.

The revised manuscript has greatly improved by including additional discussion of the model details (and shortcomings) and by significantly restructuring for clarity.

Unfortunately, from the added data and analysis of timescales, it is now clear that the timescale of remodeling in the model differs from the timescale seen in experiment by 1-2 orders of magnitude. The interpretation of the model is complicated by the fact that bulk diffusion, which the authors determine to be the slowest time-scale in experiment, is not being explicitly represented in the model.

On page 9, the authors say that Figures 3a,b and Suppl. Figures 5d,6d show the same reshaping mechanism and subsequently, make the claim that the transient states from a high concentration simulation can be interpreted as equilibrium configuration at an intermediate bulk concentrations. This claim has incomplete qualitative support with no quantitative evidence. For example, Figure 3b shows partially fused pearls while Suppl. Figure 6d does not. One explanation is that partially fused pearls in Fig. 3b are a transient state that would not be present in a series of quasi-equilibrium simulations and the reshaping mechanism from a single high concentration simulation does not properly reflect the remodeling quasi-equilibrium simulations and by extension, experiment. As such, the validity of the comparison between model and experiment in Figure 3 is highly questionable. Without further support for the claim that a high concentration simulation can be interpreted like a series of quasi-equilibrium simulations, Suppl. Videos that depict the model remodeling behavior from a single high concentration simulation should not be compared to experiment.

Reviewer #2 (Remarks to the Author):

In my opinion the authors have extensively and exhaustively responded to all reviewers comments, questions, and concerns, have performed additional theory work, and have significantly revised the manuscript text in order implement reviewers' feedback. I therefore recommend this manuscript to be accepted in its present form.

Reviewer #3 (Remarks to the Author):

The authors have addressed my requests to the best of their ability given the challenges associated with their novel experimental setup. I would like to see this novel membrane stretch-and-release system published without further delay.

Reviewer #1 (Remarks to the Author):

Recommendation: Publish after minor revisions.

The revised manuscript has greatly improved by including additional discussion of the model details (and shortcomings) and by significantly restructuring for clarity.

Unfortunately, from the added data and analysis of timescales, it is now clear that the timescale of remodeling in the model differs from the timescale seen in experiment by 1-2 orders of magnitude. The interpretation of the model is complicated by the fact that bulk diffusion, which the authors determine to be the slowest time-scale in experiment, is not being explicitly represented in the model.

In fact, we developed but not reported a model that couples bulk diffusion and the dynamics of the protein-membrane system. The coupling is only one-way since it is reasonable to assume that changes in protrusions will not affect much bulk diffusion. Thus, this model simply changes dynamically the bulk concentration to which the membrane is exposed following the bulk diffusion equation. Because of the time-scale separation with bulk diffusion being the slowest process, this model is essentially the same as the quasi-equilibrium approach that we report in the manuscript. We considered that the latter was conceptually simpler, and allowed us to perform a sequence of quasi-equilibrium snapshots at different concentrations offline, and then use the diffusion equation to map these snapshots to different time instants depending on the nominal bulk concentration.

On page 9, the authors say that Figures 3a,b and Suppl. Figures 5d,6d show the same reshaping mechanism and subsequently, make the claim that the transient states from a high concentration simulation can be interpreted as equilibrium configuration at an intermediate bulk concentrations.

Our intention was to make the claim that our experiments can be interpreted as “quasi-equilibrium states at a given dissolved protein chemical potential in the close vicinity of the SLB”. We have been more explicit now. The rationale is that both simulations (dynamical and quasi-equilibrium) produce the same mechanisms of reshaping (further detailed in the revised manuscript) and that the quasi-equilibrium simulations and the bulk diffusion equation allow us to recover the correct time-scales. We hope that our point is clearer now.

This claim has incomplete qualitative support with no quantitative evidence. For example, Figure 3b shows partially fused pearls while Suppl. Figure 6d does not. One explanation is that partially fused pearls in Fig. 3b are a transient state that would not be present in a series of quasi-equilibrium simulations and the reshaping mechanism from a single high concentration simulation does not properly reflect the remodeling quasi-equilibrium simulations and by extension, experiment.

We completely agree with the interpretation of the referee. Partially fused vesicles cannot be seen in the quasi-equilibria because they happen during the equilibration as concentration is increased, which we do not report. In the revised manuscript (page 9), we have explicitly spelled out the specific features (reshaping mechanisms) according to which the two simulation protocols are similar. We chose not to focus on the mechanism of pearl reduction (fusion or progressive disappearance) because this is not accessible experimentally.

As such, the validity of the comparison between model and experiment in Figure 3 is highly questionable. Without further support for the claim that a high concentration simulation can be interpreted like a series of quasi-equilibrium simulations, Suppl. Videos that depict the model

remodeling behavior from a single high concentration simulation should not be compared to experiment.

We think that the objection of the referee is addressed by more explicitly stating the terms of the comparison between the two simulation procedures and the experiments (the reshaping mechanisms).

Reviewer #2 (Remarks to the Author):

In my opinion the authors have extensively and exhaustively responded to all reviewers comments, questions, and concerns, have performed additional theory work, and have significantly revised the manuscript text in order implement reviewers' feedback. I therefore recommend this manuscript to be accepted in its present form.

Reviewer #3 (Remarks to the Author):

The authors have addressed my requests to the best of their ability given the challenges associated with their novel experimental setup. I would like to see this novel membrane stretch-and-release system published without further delay.